# A Distributional Approach to Uncertainty-Aware Preference Alignment Using Offline Demonstrations

**Sheng Xu[1], Bo Yue[1], Hongyuan Zha[1], Guiliang Liu[1]***
[1]School of Data Science, The Chinese University of Hong Kong, Shenzhen
{shengxu1,boyue}@link.cuhk.edu.cn
{zhahy,liuguiliang}@cuhk.edu.cn

## Abstract

Designing reward functions in Reinforcement Learning (RL) often demands significant task-specific expertise. Offline Preference-based Reinforcement Learning (PbRL) provides an effective alternative to address the complexity of reward design by learning policies from offline datasets that contain human preferences between trajectory pairs. Existing offline PbRL studies typically model a reward function by maximizing its likelihood of generating the observed human preferences. However, due to the varying number of samples within the limited dataset, less frequently compared trajectories exhibit greater uncertainty, which potentially leads to unreliable behaviors during reward and policy updates. To solve this issue, in this work, we introduce Uncertainty-Aware PbRL (UA-PbRL) to learn a distributional reward model and a risk-sensitive policy from an offline preference dataset. Our approach employs a Maximum A Posteriori (MAP) objective to update trajectory rewards and incorporates an informative prior to account for the uncertainties. Building upon this reward update, we propose a generative reward model to capture the reward distribution, utilizing the offline distributional Bellman operator and the Conditional Value-at-Risk (CVaR) metric to train a risk-sensitive policy. Experimental results demonstrate that UA-PbRL effectively identifies and avoids states with high uncertainty, facilitating risk-averse behaviors across various tasks, including robot control and language model alignment. The code is available at https://github.com/Jasonxu1225/UA-PbRL.

## 1 Introduction

In recent years, Reinforcement Learning has achieved remarkable success in addressing a variety of sequential decision problems across different domains, such as electronic games (Mnih et al., 2015; Vinyals et al., 2019; Berner et al., 2019), board games (Silver et al., 2016; 2017), and robotic manipulation (Fang et al., 2019). However, in the process of scaling these successes to real-world applications, a notable challenge is the difficulty in precisely specifying the rewards in the RL objective. Existing manually designed rewards or reward-engineering techniques cannot ensure that the learned policy aligns with the actual needs of the industry.

To develop a reliable reward function, Preference-based Reinforcement Learning (PbRL) (Knox & Stone, 2008; Christiano et al., 2017) aims to align rewards with human preferences by solving a learning-to-rank problem. In this framework, agents do not receive numerical reward signals, but are instead provided with preferences between pairs of trajectory segments, reflecting human-labeled relative judgments. PbRL has garnered considerable attention in both online (Sadigh et al., 2017; Ibarz et al., 2018; Liang et al., 2022; Park et al., 2022; Cheng et al., 2024), and offline (Shin et al., 2021; Kang et al., 2023; Zhang et al., 2024c; Choi et al., 2024; Zhan et al., 2024) settings. In the online setting, agents can interact with environments and query human experts, whereas the offline setting prohibit such interactions. In this work, we focus on the *offline PbRL* setting, where the objective is to *learn an optimal policy solely from pre-collected offline demonstrations, with preference feedback on pair-wise trajectory segments*, without any active interaction with the environment or an expert. This

---

*Corresponding author: Guiliang Liu, liuguiliang@cuhk.edu.cn.

approach is particularly promising in scenarios where collecting online trajectories and obtaining human feedback is costly and challenging (Lee et al., 2021; Shin et al., 2023).

In practice, the frequency of comparisons across different samples within the offline dataset is often highly imbalanced. While some samples may emerge with a higher ranking, they have only been compared a limited number of times. These sparsely compared trajectories introduce significant *epistemic uncertainty*, which stems from a lack of sufficient data (Smith & Gal, 2018). This uncertainty compromises the reliability of the evaluation, reducing the model's confidence in accurately assessing the value of such pairs. To ensure more reliable preference alignment, it is essential to account for the uncertainty during the alignment process. However, the Bradley-Terry model (Bradley & Terry, 1952), as a common approach to PbRL, is based on maximum likelihood estimation and lacks sensitivity to inherent uncertainties from preference datasets. Consequently, the resulting policy tends to be risk-neutral, disregarding potential safety concerns embedded in less examined decision-making scenarios.

Striving for uncertainty-aware preference alignment, several studies (Liang et al., 2022; Shin et al., 2023; Xue et al., 2023) used ensembles and Bayesian dropout to estimate the uncertainty of reward model. However, despite their empirical success, the underlying mechanism by which the estimated ensembles correlate with uncertainty in human preferences remains largely unexplained. More importantly, these approaches still rely on maximum likelihood estimation with a risk-neutral policy.

In this paper, we introduce an Uncertainty-Aware PbRL (UA-PbRL) algorithm that accounts for the uncertainty in human preferences within offline demonstrations. Specifically, we formulate a Maximum A Posteriori (MAP) objective for preference alignment, thereby incorporating signals from an informative prior to the reward updates. By interpreting preference alignment as a voting process, we intentionally select Beta distribution to implement this prior. To ensure computational tractability, we parameterize the Beta distribution with neural functions and train the model via variational inference, guided by an Evidence Lower Bound (ELBo) objective. Intuitively, this prior assigns higher probabilities to trajectories that are compared more frequently, reflecting the confidence levels associated with them. For each observed trajectory, we compute its reward using an iterative update rule, derived from the optimality conditions of our MAP objective. The reward distribution is then constructed by training a generative model conditioning on sequential state-action pairs.

To justify the effectiveness of the learned reward distribution, we train a risk-averse policy using the offline distributional Bellman operator for policy evaluation, coupled with the Conditional Value-at-Risk (CVaR) metric for policy improvement. The empirical results highlight the importance of accounting for the inherent uncertainty arising from preference demonstrations, as well as the risk-sensitive capabilities of the proposed method. In particular, UA-PbRL outperforms other baselines in pointmaze environments and robot control tasks, especially in terms of worst-case performance. Furthermore, we extended our experiments to the context of Large Language Model (LLM) alignment, demonstrating the effectiveness of UA-PbRL for LLM fine-tuning.

## 2 RELATED WORKS

**Preference-based Reinforcement Learning.** PbRL considers aligning policies with human preferences, eliminating the requirement for explicit reward signals (MacGlashan et al., 2017; Warnell et al., 2018). Previous studies have successfully combined PbRL to deep RL agent (Christiano et al., 2017) and high-dimensional image space (Ibarz et al., 2018). Building on this foundation, several methods (Lee et al., 2021; Liang et al., 2022; Park et al., 2022) further improve the sample efficiency by incorporating techniques like pre-training and relabeling. Recent studies include learning from few preference labels (Liu et al., 2022) and meta-learning (Hejna III & Sadigh, 2023). Some other works introduce hindsight techniques (Verma & Metcalf, 2024; Singh et al., 2024). Recently, offline PbRL has gained attention for optimizing policies with an offline dataset, addressing safety and sample efficiency concerns. Existing offline PbRL algorithms primarily utilize the Bradley-Terry model (Bradley & Terry, 1952) to model the likelihood of human preference based on reward signals (Shin et al., 2021; Kim et al., 2023; Choi et al., 2024; Zhang et al., 2024c). However, such a maximum likelihood method is insensitive to the underlying confidence in human preferences (Newman, 2023). Another line of research bypasses the need for learning a reward model by directly aligning policies with human preferences (An et al., 2023; Kang et al., 2023; Hejna & Sadigh, 2023; Hejna et al., 2024), while these approaches similarly neglect the uncertainty. Effectively handling this uncertainty and deriving risk-sensitive policies remains a critical challenge (Casper et al., 2023).

**Distributional Reinforcement Learning.** While the majority of RL research focuses on maximizing the expected cumulative rewards, Bellemare et al. (2017) introduces a distributional perspective on RL, utilizing the distributional Bellman operator for value function updates. Such distributional value functions are sensitive to the uncertainty in the environment dynamics (Mavrin et al., 2019), enabling the formulation of risk-sensitive policies (Lim & Malik, 2022; Keramati et al., 2020) and better controlling performance (Bellemare et al., 2023). Some previous studies propose utilizing categorical distribution (Bellemare et al., 2017; Sui et al., 2023), quantile functions (Dabney et al., 2018b;a; Zhang & Yao, 2019; Zhou et al., 2020; 2021; Luo et al., 2022) and diffusion models (Wu et al., 2023) for representing and updating the distributional value function. In this work, we utilize quantile functions since the statistical benefit of quantile regression is most well-understood (Rowland et al., 2023). Some recent studies extend distributional RL to offline learning (Ma et al., 2021), multi-dimensional rewards (Zhang et al., 2021), and multi-agent control (Hu et al., 2022; Sun et al., 2021). However, none of the previous works have considered PbRL from a distributional perspective.

## 3 PROBLEM FORMULATION

**Markov Decision Process (MDP).** The agent optimizes the control policy under a Markov Decision Process (MDP) $\mathcal{M} = (\mathcal{S}, \mathcal{A}, R, p_\mathcal{T}, \mu_0, \gamma)$, where 1) $\mathcal{S}$ and $\mathcal{A}$ denote the state and action spaces, 2) $p_\mathcal{T} : \mathcal{S} \times \mathcal{A} \to \Delta^\mathcal{S}$ denotes the stochastic transition function, where $\Delta^\mathcal{S}$ is the simplex over $\mathcal{S}$, 3) $R : \mathcal{S} \times \mathcal{A} \to [R_{\min}, R_{\max}]$ denotes the reward function, 4) $\mu_0 \in \Delta^\mathcal{S}$ denotes the initial state distribution, and 5) $\gamma \in (0, 1]$ is the discounting factor. For brevity, we use $\mathcal{M}_{/R}$ to denote the MDP without knowing the reward. In this work, we mainly study the episodic MDPs where the planning stops at a terminating state, and the corresponding terminating time is denoted as $T \in (0, \infty)$.

**Risk-Sensitive Reinforcement Learning.** Under an MDP, the objective is to learn a policy $\pi : \mathcal{S} \to \mathcal{A}$, which optimizes the following objective:

$$\pi = \arg\max_\pi \rho_{\mu_0, p_\mathcal{T}, \pi}^\alpha [\sum_{t=0}^T R(s_t, a_t)].$$

Instead of optimizing the risk-neutral expected cumulative rewards, we consider a risk-sensitive measure $\rho_{\mu_0, p_\mathcal{T}, \pi}^\alpha$, where the confidence level $\alpha < 1$. Specifically, by implementing $\pi$ under the MDP $\mathcal{M}$, we generate a trajectory $\tau \in (\mathcal{S} \times \mathcal{A})^T$. The corresponding trajectory-generating probability can be defined as $p^\pi(\tau) = \mu_0(s_0) \prod_{t=0}^{T-1} \pi(a_t|s_t) p_\mathcal{T}(s_{t+1}|s_t, a_t)$. We define the corresponding risk envelope $\mathcal{U}_\alpha^\pi = \{\zeta_\alpha : \Gamma \to [0, \frac{1}{\alpha}] \mid \sum_{\tau \in \Gamma} \zeta(\tau) p^\pi(\tau) = 1\}$ to be a compact, convex, and bounded set, based on which the risk measure can be induced by the distorted probability distribution $p_\zeta^\pi = \zeta \cdot p^\pi$. In this work, we study the CVaR such that $\rho_\alpha^\pi [\sum_{t=0}^T \gamma^t R_t] = \sup_{\zeta_\alpha \in \mathcal{U}_\alpha^\pi} \mathbb{E}_{\tau \sim p^\pi} [\zeta_\alpha(\tau) \sum_{t=0}^T \gamma^t R_t]$ due to its time consistency and convexity (Rockafellar et al., 2000).

**Uncertainty-Aware Preference-based Reinforcement Learning (UA-PbRL).** Previous research on PbRL typically learns a deterministic reward function (Christiano et al., 2017; Kim et al., 2023) under a maximum likelihood objective. However, in real-world applications, the frequency of comparisons among different samples within the offline dataset is often highly imbalanced, which can significantly influence the safety and reliability of downstream control tasks. Intuitively, if a trajectory and its counterparts have only been assessed a few times, the corresponding preference signals should have large uncertainty. This uncertainty is closely linked to the model's confidence: the greater the uncertainty, the lower the confidence in its predictions (Smith & Gal, 2018; Mena et al., 2021).

To better accommodate the underlying uncertainty in human preference datasets, we study an uncertainty-aware objective for achieving UA-PbRL. Specifically, we capture the uncertainty by learning a distributional reward model $f_\varphi^r : \mathcal{S} \times \mathcal{A} \to \Delta^{[R_{\min}, R_{\max}]}$ (Section 4), and incorporate these uncertainty signals into policy learning by utilizing the offline distributional policy evaluation and the risk-averse policy improvement (Section 5). To more accurately reflect the inherent uncertainty, we assume the offline preference dataset is inherently constructed as follows:

**Assumption 3.1.** *(Offline Preference Dataset). The preference dataset is constructed with a set of policies $\Pi = \{\pi_1, \ldots, \pi_L\}$ (with size $L$), where each policy exhibits distinct behaviors. For each candidate policy $\pi_l \in \Pi$, we generate $N_l$ trajectories in the environment $\mathcal{M}_{/R}$. We uniformly sample a pair of trajectories $(\tau, \tau')$ from a total number of $N_l \cdot L$ generated trajectories. Humans express preferences by mapping $(\tau, \tau')$ to $(\tau^i, \tau^j)$ where $\tau^i$ ranks higher than $\tau^j$. We repeat the sampling and mapping process until we generate the dataset $\mathcal{D}$.*

This data-collection paradigm is common in applications where publicly available or structured data is lacking, such as robot control commands for embodied AI or event-related dialogue data for chatbots. Even in tasks with abundant demonstration data, such as autonomous driving, the dataset is typically collected from various agents (e.g., human drivers), each following their own distinct policies.

**Imbalanced Preference Dataset.** During the trajectory generation process, under different contexts, agents' policies tend to behave differently, exhibiting various levels of diversity in their movements. For example, as illustrated in Figure 1 (left), policy $\pi_2$ chooses to move right and exhibits more diverse behaviors, *potentially due to the inherent stochasticity in the policy itself or the dynamics of the environment*. This diversity leads to the generation of a broader range of trajectories $\tau_2^1, \ldots, \tau_2^{N_2}$, compared to the more deterministic trajectories $\tau_1^1, \ldots, \tau_1^{N_1}$ from $\pi_1$. Due to this diversity, a specific trajectory such as $\tau_2'$ is less likely to be selected during the uniform sampling because of its scarcity. Consequently, these trajectories will undergo fewer human comparisons, leading to lower confidence for the reward model in assessing their true values.

Such a phenomenon is common in practice. For example, a risk-seeking operator may guide the robot into unfamiliar environments. Given the partial observability of sensors and the presence of unpredictable obstacles, the resulting policy tends to exhibit greater diversity compared to those in more routine, less challenging environments. The scarcity of the resulting control trajectories must be considered by assigning them lower confidence in the ranking or evaluation process.

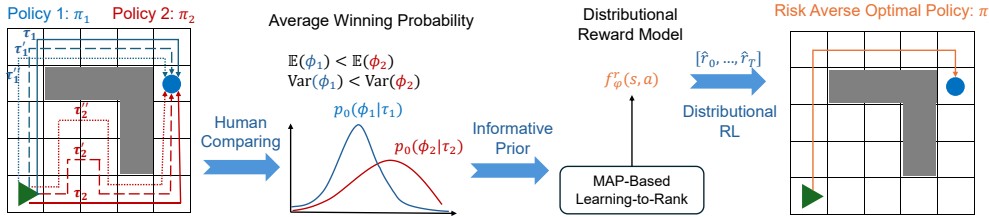

Figure 1: An illustrative example of UA-PbRL. In this map, humans uniformly sample and compare pairs of trajectories generated by policies $\pi_1$ and $\pi_2$. The trajectory $\tau_2$ is shorter in expectation, so $\tau_2$ is more likely to outperform the average trajectory compared to the longer $\tau_1$. However, $\pi_2$ generates more diverse trajectories, resulting in fewer human comparisons and thus greater uncertainty for each trajectory. Such uncertainty is captured by the distributional reward model, which steers a risk-averse policy to navigate through a longer but more reliable path on the top-left map.

Based on the dataset $\mathcal{D}$, we consider the **Offline PbRL** problem, where the agent has access solely to an offline dataset that records labeled trajectories instead of interacting directly with environments.

## 4 LEARNING GENERATIVE REWARD MODEL FROM HUMAN PREFERENCE

In this section, we present our approach to learning a distributional reward model by proposing: 1) an MAP objective for inferring rewards from human preference (Sec. 4.1), 2) an informative Beta prior for modeling uncertainty (Sec. 4.2), and 3) the method of learning generative rewards (Sec. 4.3).

### 4.1 MAXIMUM A POSTERIORI OBJECTIVE FOR REWARD INFERENCE

Previous PbRL algorithms commonly utilize the Bradley-Terry model (Bradley & Terry, 1952) to represent the log-likelihood of generating human preferences with the reward function:

$$\mathcal{L}(\varphi, \mathcal{D}) = \frac{1}{|\mathcal{D}|} \sum_{(\tau^i, \tau^j) \in \mathcal{D}} \omega^{\tau^i, \tau^j} \log \frac{e^{[r_\varphi(\tau^i)]}}{e^{[r_\varphi(\tau^i)]} + e^{[r_\varphi(\tau^j)]}}, \tag{1}$$

where $r_\varphi(\tau) = \sum_{t=0}^{T} \gamma^t r_\varphi(s_t, a_t)$ denotes the trajectory segment rewards parameterized by $\varphi$, $(\tau^i, \tau^j)$ denotes a pair of trajectories where $\tau^i$ ranks higher than $\tau^j$, and $\omega^{\tau^i, \tau^j}$ denotes the frequency of such pairs appearing in $\mathcal{D}$. The maximum likelihood objective implicitly imposes a uniform prior for $r_\varphi(\tau)$ such that $p_0(r_\varphi(\tau)) = 1/(\frac{R_{\max}}{1-\gamma} - \frac{R_{\min}}{1-\gamma})$. It places a vanishing fraction of its weight on arbitrarily large values , which causes divergence in the reward function's parameters (Newman, 2023).

In a learning-to-rank problem, the winning probability that trajectory $\tau^i$ wins against $\tau^j$ remains unchanged under multiplication of all the $e^{r_\varphi(\tau)}$ by any constant factor. To derive a more useful prior for the reward function, we enforce the geometric mean strength to be one ($\prod_n e^{r_\varphi(\tau^n)} = 1$,

i.e., $\sum_n r_\varphi(\tau^n) = 0$) and thus the probability of a player with strength $e^{r_\varphi(\tau)}$ winning against the average player (whose strength $e^{r_\varphi(\bar{\tau})} = 1$) is $\phi(\tau) = e^{r_\varphi(\tau)}/(e^{r_\varphi(\tau)} + 1)$. Consequently, the reward represents the log odds of defeating the average trajectory. For brevity, we use $\phi$ as a shorthand of $\phi(\tau)$ to denote such probability. Then the prior on the reward function can be defined as:

$$p_0(r_\varphi(\tau)) = p_0(\phi)\frac{d\phi}{dr_\varphi(\tau)} = p_0(\phi)\frac{d\phi}{de^{r_\varphi(\tau)}}\frac{de^{r_\varphi(\tau)}}{dr_\varphi(\tau)} = p_0(\phi)\frac{e^{r_\varphi(\tau)}}{(e^{r_\varphi(\tau)} + 1)^2}, \qquad (2)$$

where the first two equations can be derived by applying the chain rule. $p_0(\phi)$ is the prior, which can follow different representations. This update enables the definition of an MAP objective:

$$(3)$$

$$p(r_\varphi(\tau)|\mathcal{D}) \propto p(\mathcal{D}|r_\varphi(\tau))p_0(r_\varphi(\tau)) = \prod_{(\tau^i,\tau^j)\in\mathcal{D}}\left[\frac{e^{[r_\varphi(\tau^i)]}}{e^{[r_\varphi(\tau^i)]} + e^{[r_\varphi(\tau^j)]}}\right]^{\omega^{\tau^i,\tau^j}}\prod_{\tau^i}p_0(\phi)\frac{e^{[r_\varphi(\tau^i)]}}{(e^{[r_\varphi(\tau^i)]} + 1)^2}.$$

Instead of maximizing the likelihood, maximizing this posterior probability can integrate prior knowledge and regularize the reward values, preventing them from diverging.

An essential prerequisite for implementing UA-PbRL with this MAP objective is the construction of an informative prior, $p_0(\phi)$. This prior incorporates the inherent uncertainty of human preferences into the reward learning process. We introduce the estimation of $p_0(\phi)$ in the following section.

## 4.2 LEARNING INFORMATIVE BETA PRIOR

In this study, we learn a Beta distribution as the informative prior from $\mathcal{D}$, i.e., $p_0(\phi|\mathcal{D}) = \text{Beta}(\alpha, \beta)$, since 1) the Beta distribution is the conjugate prior for the Bernoulli distribution, facilitating the update of our beliefs with new evidence; 2) the parameters $\alpha$ and $\beta$ of the Beta distribution can be effectively interpreted as representing the count of positive and negative human feedback, respectively, where "positive" refers to being preferred, while "negative" indicates not preferred. For a trajectory $\tau$, as the number of such "votes" increases, our confidence in the inferred probability improves, resulting in a more precise (or "sharper") distribution. This approach enables quantitatively incorporating the confidence level of the Bernoulli probability estimation into our model. Building on this, we present a discrete approach to learning the Beta distribution using statistics, detailed in Appendix B.4.

To learn the distribution in continuous spaces, we propose the variational inference approach to approximate $p_0(\phi|\mathcal{D})$ by estimating the approximate posterior $q_\psi(\phi|\mathcal{D})$ (i.e., $p_0(\phi|\mathcal{D}) \simeq q_\psi(\phi|\mathcal{D})$). The goal of our variational inference approach is to learn an approximate posterior distribution $q_\psi(\phi|\mathcal{D})$ by minimizing the Kullback–Leibler (KL) divergence $D_{kl}(q_\psi(\phi|\mathcal{D})\|p(\phi|\mathcal{D}))$:

$$D_{kl}\Big(q_\psi(\phi|\mathcal{D})\|p(\phi|\mathcal{D})\Big) = -\mathbb{E}_{q_\psi}\Big[\log p(\mathcal{D}|\phi)\Big] + D_{kl}\Big[q_\psi(\phi|\mathcal{D})\|p(\phi)\Big] + \log\Big[p(\mathcal{D})\Big]. \quad (4)$$

Minimizing the above objective is equivalent to maximizing the Evidence Lower Bound (ELBo) $\log[p(\mathcal{D})] - D_{kl}(q_\psi(\phi|\mathcal{D})\|p(\phi|\mathcal{D}))$. By following Equation (4), ELBo can be represented as:

$$\mathbb{E}_{q_\psi}\Big[\log p(\mathcal{D}|\phi)\Big] - D_{kl}\Big[q_\psi(\phi|\mathcal{D})\|p(\phi)\Big]. \qquad (5)$$

The corresponding trajectory-wise objective can be reinterpreted as follows:

$$\max_\psi \mathbb{E}_\tau\Big[\mathbb{E}_{q_\psi,(\tau,\tau')\in\mathcal{D}}[\log\phi(\tau)] - \mathbb{E}_{q_\psi,(\tau',\tau)\in\mathcal{D}}[\log\phi(\tau)] - D_{kl}[q_\psi(\phi|\tau)\|p(\phi)]\Big], \qquad (6)$$

where 1) $q_\psi(\phi|\tau) = \text{Beta}(\alpha_\tau, \beta_\tau)$, where $[\alpha_\tau, \beta_\tau] = f_\psi^{\text{Beta}}(\tau)$, and $f_\psi^{\text{Beta}}$ denotes a neural network parameterized by $\psi$, 2) $p(\phi) = \text{Beta}(\alpha_0, \beta_0)$ where $\alpha_0, \beta_0$ defines our initial belief (hyperparameters, check Appendix B.3 for details), and 3) $\phi(\tau)$ denotes the Bernoulli probability that $\tau$ ranks higher than $\tau'$. Since both the posterior distribution $q_\psi(\phi|\tau)$ and the prior distribution $p(\phi)$ are Beta-distributed, we represent the KL divergence term by following the Dirichlet VAE (Joo et al., 2020):

$$D_{kl}[q_\psi(\phi|\tau)\|p(\phi)] = \log\left(\frac{\Gamma(\alpha_\tau + \beta_\tau)}{\Gamma(\alpha_0 + \beta_0)}\right) + \log\left(\frac{\Gamma(\alpha_0)\Gamma(\beta_0)}{\Gamma(\alpha_\tau)\Gamma(\beta_\tau)}\right) \qquad (7)$$
$$+ (\alpha_\tau - \alpha_0)\Big[\Psi(\alpha_\tau) - \Psi(\alpha_\tau + \beta_\tau)\Big] + (\beta_\tau - \beta_0)\Big[\Psi(\beta_\tau) - \Psi(\alpha_\tau + \beta_\tau)\Big],$$

where 1) $[\alpha_0, \beta_0]$ and $[\alpha_\tau, \beta_\tau]$ are parameters from the prior and the posterior functions, and 2) $\Gamma$ and $\Psi$ denote the gamma and the digamma functions.

### 4.3 LEARNING GENERATIVE REWARD MODEL

In this work, we leverage a conditional generative model $f_\varphi^r$ to represent the joint distribution of the step-wise rewards in a trajectory, i.e., $\hat{r}(\tau) \sim p(r|\tau) = f_\varphi^r(\tau)$, where $\hat{r}(\tau) = \sum_t \gamma^t \hat{r}_t$ denotes the trajectory reward. To enable efficient estimation, we derive an iterative update rule based on the MAP objective and estimated Beta prior. Specifically, each time we sample rewards from the joint distribution, and then update these rewards based on the following iterative update rule:

$$\hat{r}^{k+1}(\tau) = \log \frac{\alpha_\tau/[e^{\hat{r}^k(\tau)} + 1] + \sum_j \omega^{\tau,\tau^j} e^{\hat{r}^k(\tau^j)}/[e^{\hat{r}^k(\tau)} + e^{\hat{r}^k(\tau^j)}]}{\beta_\tau/[e^{\hat{r}^k(\tau)} + 1] + \sum_i \omega^{\tau^i,\tau}/[e^{\hat{r}^k(\tau)} + e^{\hat{r}^k(\tau^i)}]}, \tag{8}$$

where $\omega^{\tau,\tau^j}$ and $\omega^{\tau^j,\tau}$ are calculated based on the dataset $\mathcal{D}$. The design of this iterative update rule is based on the following theorem:

**Theorem 4.1.** *Let the informative prior $p_0(\phi)$ be a beta distribution $Beta(\alpha, \beta)$ and $e^{r(\tau)}$ be the strength of a trajectory segment $\tau$. Assuming the geometric mean strength to be 1, i.e., $\prod e^{r(\tau)} = 1$, the iteration of Equation (8) will converge to the maximum of its MAP objective (i.e., Equation (3) with $p_0(\phi)$ the Beta prior), from any starting point, whenever a maximum exists.*

The proof can be found in Appendix A. To train the generative reward model, we 1) sample rewards from this model, 2) refine these rewards based on Equation (8) under the guidance of human preferences, and 3) update the reward model by fitting it to the updated rewards (see Algorithm 1).

**Model Implementation.** In this work, we implement $f_\varphi^r$ by a *distributional reward transformer* parameterized by $\varphi$. For each trajectory $\tau$, we sample the initial step-wise rewards $[r_0^0, \ldots, r_T^0]$ from this reward model $f_\varphi^r(\tau)$ and calculate $\hat{r}^0(\tau) = \sum_{t=0}^T \gamma^t \hat{r}_t^0$. By utilizing the reward update process (Equation (8)), we compute the updated segment rewards $\hat{r}^K(\tau)$ after $K$ iterations, which approach the actual MAP values due to Theorem 4.1. The corresponding loss function can be modeled as:

$$\min_\varphi \mathbb{E}_\mathcal{D} \left[ [(\hat{r}^K(\tau) - \hat{r}^0(\tau)]^2 \right] \text{ where } \hat{r}_t^0 \sim \mathcal{N}(\mu_t, \sigma_t). \tag{9}$$

Here $\hat{r}_t^0$ is sampled from a Gaussian distribution parameterized by mean $\mu_t$ and variance $\sigma_t^2$ like in (Liu et al., 2024) with dropout layers (Srivastava et al., 2014). To derive tractable gradients, we apply the reparameterization trick to generate $\hat{r}_t^0 = \mu_t + \sigma_t \cdot \epsilon$ where $\epsilon$ denotes samples from standard Gaussian distribution. Both $\mu_t$ and $\sigma_t$ are the predictions of a causal transformer such that:

$$[(\mu_t, \sigma_t)_{t=0}^T] = \text{CausalTransformer}(s_0, a_0, \ldots, s_T, a_T). \tag{10}$$

The detailed implementation can be found in Appendix B.1.

## 5 RISK-SENSITIVE POLICY OPTIMIZATION

In this section, we introduce the approach to learning risk-sensitive policy that aligns with the inherent uncertainty in human preferences. Specifically, we employ the distributional Bellman operator to model the distribution of discounted cumulative rewards from the offline dataset (Sec. 5.1). Given the estimated value distribution, we carry out policy improvement by maximizing the CVaR (Sec. 5.2).

### 5.1 OFFLINE DISTRIBUTIONAL POLICY EVALUATION

To enable distributional policy evaluation, we incorporate the learned reward generator $f_\varphi^r$ to the original MDP $\mathcal{M}_{/R}$ without knowing the ground-truth reward. The resulting running environment is denoted as $\mathcal{M}_{/R} \cup f_\varphi^r$. For brevity, we denote it as $\widehat{\mathcal{M}}$.

Given a policy $\pi$, our goal is to learn a distributional action-value function $Z_{\widehat{\mathcal{M}}}^\pi(s, a)$ to estimate the distribution of discounted cumulative reward $\sum_{t=0}^\infty \gamma^t R(s_t, a_t)$ where the initial state-action pair $(s_0, a_0)$ is based on an offline dataset $\mathcal{D}$. We represent the distribution of $Z_{\widehat{\mathcal{M}}}^\pi$ by a uniform mixture of supporting quantiles such that $Z_{\widehat{\mathcal{M}}}^\pi(s_t, a_t) = \mathbb{E}_{\xi \sim U(0,1)}[\delta_{\theta_\xi(s_t, a_t)}]$, where $\theta_\xi$ estimates the quantile at the quantile level $\xi$ and $\delta_{\theta_\xi}$ denotes a Dirac distribution at $\theta_\xi$.

To implement offline update for the model parameters $\theta$, we utilize the following Conservative Distribution Evaluation (CDE) objective (Ma et al., 2021):

$$\min_\theta \ \mathcal{L}_{TD}(\theta) + \lambda \mathbb{E}_{\xi \sim U[0,1]} \left[ \mathbb{E}_{s \sim \mathcal{D}} (\log \sum_a \exp \theta_\xi(s, a)) - \mathbb{E}_{(s,a) \sim \mathcal{D}} (\theta_\xi(s, a)) \right], \tag{11}$$

$$\mathcal{L}_{TD} = \mathbb{E}_{\mathcal{D}}\Big[\mathbb{E}_{(\xi,\xi')\sim U(0,1)}\left[\rho_\kappa^\xi\left(\hat{r}_t + \gamma\theta_{\xi'}(s_{t+1}, a_{t+1}) - \theta_\xi(s_t, a_t)\right)\right]\Big], \tag{12}$$

where 1) $\lambda$ is the penalty weight, 2) $\rho_\kappa^\xi$ is the $\xi$-Huber quantile regression loss at threshold $\kappa$ (Huber, 1964), and 3) $(s_t, a_t, s_{t+1}, a_{t+1})$ is uniformly sampled from the trajectories in the dataset $\mathcal{D}$ while $\hat{r}_t$ is sampled from our reward generator $f_\varphi^r$.

## 5.2 RISK-AVERSE POLICY IMPROVEMENT

To better handle the underlying uncertainty in human preference, we adopt risk-averse policy updates by maximizing the estimate of CVaR within return distributions. However, Lim & Malik (2022) indicates that directly integrating CVaR-based policy improvement with distributional policy evaluation does not necessarily guarantee convergence to the optimal policy. To overcome this issue, we utilize the following distributional policy improvement objective for static CVaR (Lim & Malik, 2022):

$$\pi(a_{t+1}|s_{t+1}) = \arg\max_{a_{t+1}} \mathbb{E}_{\xi\sim U[0,1]}\left[-(q(s_{t+1}) - \theta_\xi(s_{t+1}, a_{t+1}))^+\right], \tag{13}$$

where $q(s_{t+1}) = (q(s_t) - r_t)/\gamma$ keeps track of the reward history with the initial value of $q^\alpha = \mathbb{E}_{a\sim\pi}\left[F_{Z^\pi(s,a)}^{-1}(\alpha)\right]$, where $F_{Z^\pi(s,a)}^{-1}$ denotes the inverse cumulative density function of distribution $Z^\pi(s, a)$. As is shown in (Bäuerle & Ott, 2011), $\pi$ converges to the optimal static CVaR policy by iteratively calculating $q$ and updating $\pi$ under an MDP whose state space is augmented by $q$ (i.e., $\tilde{s} = (s, q) \in \mathcal{S} \times \mathcal{R}$). Intuitively, $q$ is a moving threshold keeping track of the accumulated rewards until the current time step.

**Practical Implementation.** The complete UA-PbRL algorithm is presented in Algorithm 1.

---

**Algorithm 1:** Uncertainty-Aware Preference-based Reinforcement Learning (UA-PbRL)

---

**Input:** The preference dataset $\mathcal{D}$, reward learning epochs $N$, maximum iterations $K$
Initialize reward model $f_\varphi^r$, Beta model $f_\psi^{\text{Beta}}$, action-value model $Z_{\widehat{\mathcal{M}}}^\pi(s, a)$, and policy $\pi(a|s)$ ;
Build a buffer $\mathcal{B}_\tau$ that records all the recorded trajectories in $\mathcal{D}$;
Update the Beta prior model $f_\psi^{\text{Beta}}$ with the objective (6) until convergence;
**for** $n = 1, 2, \cdots, N$ **do**                // Distributional Reward Learning.
    **for** $\tau \in \mathcal{B}_\tau$ **do**
        Sample rewards $[\hat{r}_0, \ldots, \hat{r}_T] \sim f_\varphi^r(\tau)$ and calculate $\hat{r}(\tau) = \sum_{t=0}^T \gamma^t \hat{r}_t$;
        Estimate the beta prior $[\hat{\alpha}(\tau), \hat{\beta}(\tau)] = f_\psi^{\text{Beta}}(\tau)$;
        Calculate the updated $\hat{r}^K(\tau)$ with the objective (8);
        Update the reward model $f_\varphi^r$ with the objective (9).
    **end**
**end**
**for** $\tau \in \mathcal{B}_\tau$ **do**                   // Risk-Sensitive Policy Optimization.
    Sample step-wise rewards $[\hat{r}_0, \ldots, \hat{r}_T] \sim f_\varphi^r(\tau)$ for the trajectory $\tau$;
    Update the distributional action-value function $Z_{\widehat{\mathcal{M}}}^\pi(s, a)$ with the objective (11);
    Update the policy model $\pi(a|s)$ with the objective (13);
**end**

---

## 6 EMPIRICAL EVALUATION

In the empirical study, we start by illustrating the learned distributional reward model in discrete Gridworld environments (Section 6.1). Next, we construct three Risky PointMaze environments and empirically evaluate the effectiveness of the proposed UA-PbRL algorithm with trajectory visualization (Section 6.2). To assess performance in more challenging settings, we also examine two complex robot navigation tasks (Section 6.3). Lastly, we extend the experiments to explore the application in Large Language Model alignment (Section 6.4).

**Experiment Settings.** Our experiments primarily utilize the public platform Uni-RLHF (Yuan et al., 2024), which is tailored for offline PbRL. Additionally, to accommodate the underlying uncertainties during the preference learning process, we introduce risky regions by incorporating noise into transitions within them, during both trajectory collecting and policy evaluating processes. We create the offline dataset by uniformly sampling from the expert policies trained online and then generating

preferences based on their true rewards and risky steps. Consequently, *the trajectories collected within the risky region are more diverse, leading to fewer comparisons and greater uncertainty*. Please check Appendix B.2 for more details. For fairness, all methods used in robot control tasks are built upon the Preference Transformer (PT) (Kim et al., 2023) for reward learning and Conservative Q Learning (CQL) (Kumar et al., 2020) for policy optimization.

By following Ma et al. (2021), we evaluate each approach using 100 test episodes by reporting both the mean and $\text{CVaR}_{0.1}$ (i.e., the average over the worst 10 episodes) metrics, including: 1) *episodic rewards*, which calculate the cumulative rewards within an episode, and 2) *episodic violations*, which aggregate the total number of time steps spent inside the risky region. Each experiment is repeated with four random seeds, and the results are presented with mean $\pm$ standard deviation (std).

## 6.1 REWARD VISUALIZATION IN GRIDWORLD

In this experiment, we construct a Gridworld environment to better illustrate the case previously described in Figure 1. As shown in the left plot of Figure 2, the objective for the agent is to navigate from an initial position (green arrow) to a specified target (blue circle) while avoiding the walls (grey blocks). Within the bottom-right area (red markers), the environment demonstrates a degree of stochasticity, where, with specific probabilities ($p = 0.1$), it receives a random action instead of the agent's intended action (refer to Appendix C.1 for more details). Intuitively, the trajectories $\tau_2$ generated by $\pi_2$ (passing through right-bottom) exhibit higher rewards in expectation. To accommodate this situation, we assign greater preference to $\tau_2$ by setting the expected chance of observing the preference that $\tau_2$ ranks higher than $\tau_1$ to be $p = 0.6$.

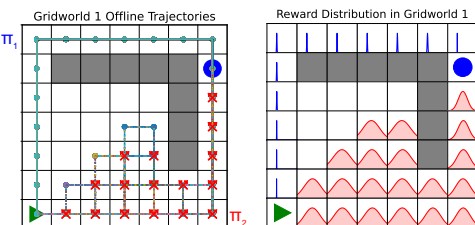

Figure 2: (Left) The Gridworld environment and offline trajectories. (Right) The learned risk-sensitive reward distributions by our method. Please refer to Figure 8 in Appendix D.1 for the mean and standard deviation values of each distribution and the results in the remaining two settings.

The right plot of Figure 2 illustrates the learned reward distributions at each state, where we utilize the blue and red colors to represent the rewards at risk-averse and risky regions, respectively. We find that the distributional reward model successfully captures the underlying uncertainty within the offline dataset in the sense that the rewards in risky regions exhibit a larger expectation but a higher variance. This leads to the result that the generated risk-averse policy avoids the risky area and navigates through the top-left map. Additionally, we also construct two distinct Gridworlds and illustrate the corresponding rewards. Please check Figure 8 in Appendix D.1 for complete results. We also visualize the results with preference strengths of $p = 0.7$ and $0.8$ in Figures 9, 10, and 11.

## 6.2 MODEL PERFORMANCE IN RISKY POINTMAZE

**Task Description.** In this experiment, we extend to the continuous domain by constructing three PointMaze environments, as shown in Figure 3. In the risky regions denoted by red markers, the environmental transitions are influenced by additional Gaussian noise calculated such that $p_{\mathcal{T}}(s_{t+1}|s_t, a_t) = f(s_t, a_t) + \mathcal{N}(\mu_1, \sigma_1)$, where $f(\cdot)$ denotes the original transition function. Please check Appendix C.2 for more environmental details.

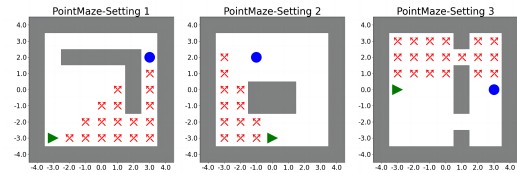

Figure 3: Three Risky PointMaze settings.

**Comparison Methods.** Besides our **UA-PbRL** algorithm that learns a distributional reward model with a risk-averse policy, the following baselines are compared: 1) *regular PbRL* (**PbRL**) Christiano et al. (2017) that learns a reward model through the Maximum Likelihood Estimation (MLE) objective, 2) *Ensemble Neutral PbRL* (**EN-PbRL**) (Liang et al., 2022) that learns an ensemble of reward functions (we use five ensembles) with risk-neutral policy optimization. 3) *Ensemble Risk-sensitive PbRL* (**ERS-PbRL**) that replaces the neutral objective in EN-PbRL with a risk-averse one (Ma et al., 2021). 4) *Contrastive Preference Learning* (**CPL**) (Hejna et al., 2024) that learns policies directly from preferences without reward learning.

**Results Analysis.** Table 1 shows the evaluation performance, with the best results in each setting (highest rewards or lowest violations) highlighted in bold. Please check Figures 12 and 13 in

Table 1: Evaluation results in three Risky PointMaze settings. Each value is reported as the mean $\pm$ standard deviation (std) calculated from 100 episodes and 4 seeds.

| | Method | | CPL | PbRL | EN-PbRL | ERS-PbRL | UA-PbRL (ours) |
|---|---|---|---|---|---|---|---|
| PointMaze Setting 1 | Rewards ↑ | Mean | **47.6 ± 16.4** | 30.8 ± 27.4 | 40.7 ± 18.6 | 25.0 ± 4.1 | 32.6 ± 33.5 |
| | | $\text{CVaR}_{0.1}$ | -30.3 ± 44.6 | -60.0 ± 0.0 | -60.0 ± 0.0 | -60.0 ± 0.0 | **-16.6 ± 58.9** |
| | Violations ↓ | Mean | 310.5 ± 32.4 | 272.7 ± 20.2 | 265.8 ± 12.2 | 172.3 ± 19.7 | **86.1 ± 86.4** |
| | | $\text{CVaR}_{0.1}$ | 494.9 ± 101.6 | 450.7 ± 23.6 | 452.4 ± 46.0 | 273.8 ± 87.2 | **187.3 ± 134.4** |
| PointMaze Setting 2 | Rewards ↑ | Mean | 38.7 ± 27.1 | 63.2 ± 5.0 | 64.0 ± 6.4 | **65.8 ± 5.2** | 65.4 ± 11.2 |
| | | $\text{CVaR}_{0.1}$ | -15.7 ± 44.3 | 42.3 ± 16.7 | 43.0 ± 17.3 | 48.3 ± 8.0 | **53.1 ± 10.5** |
| | Violations ↓ | Mean | 172.3 ± 12.5 | 121.2 ± 5.9 | 125.9 ± 15.3 | 98.0 ± 16.1 | **5.2 ± 7.3** |
| | | $\text{CVaR}_{0.1}$ | 225.3 ± 53.6 | 150.3 ± 23.4 | 163.9 ± 31.9 | 147.9 ± 8.8 | **50.8 ± 71.7** |
| PointMaze Setting 3 | Rewards ↑ | Mean | 38.3 ± 47.3 | 64.2 ± 11.9 | 68.6 ± 7.5 | 67.7 ± 11.0 | **70.3 ± 15.4** |
| | | $\text{CVaR}_{0.1}$ | -1.5 ± 64.6 | 22.7 ± 51.7 | 41.8 ± 16.7 | 19.2 ± 68.5 | **48.9 ± 55.1** |
| | Violations ↓ | Mean | 83.6 ± 123.6 | 71.1 ± 71.7 | 107.1 ± 71.2 | 53.7 ± 47.0 | **27.7 ± 33.1** |
| | | $\text{CVaR}_{0.1}$ | 205.0 ± 235.6 | 167.5 ± 120.3 | 185.0 ± 61.5 | 143.2 ± 68.1 | **117.7 ± 96.4** |

Appendix D.2 for evaluation results during the complete training phase. The results show that UA-PbRL consistently outperforms other methods with higher $\text{CVaR}_{0.1}$ rewards and fewer violations in both mean and $\text{CVaR}_{0.1}$ metrics. This underscores that the risk-averse policy in UA-PbRL can avoid passing through highly uncertain regions. When it comes to the mean rewards, UA-PbRL still achieves compatible performance due to its superior $\text{CVaR}_{0.1}$ performance. We also find that ERS-PbRL and CPL sometimes achieve higher mean rewards than UA-PbRL. This is because the two methods do not acknowledge the risky regions and solely pursue expected cumulative rewards, resulting in traversing through risky areas with occasional successes.

**Results Visualization.** Figure 4 illustrates 10 evaluation rollouts from PbRL and UA-PbRL in the first setting of Risky PointMaze (check Figure 14 in Appendix D.2 for complete results). We find that UA-PbRL drives a risk-averse policy that navigates to the longer but less stochastic path. By contrast, the traditional PbRL method struggles to perceive such uncertainties and tends to navigate through the risky region directly, where the noisy transition occasionally induces unsafe movements, leading to its poor $\text{CVaR}_{0.1}$ performance.

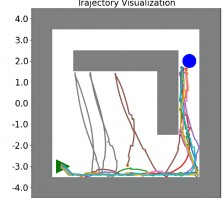 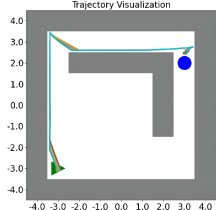

Figure 4: The trajectories generated by PbRL (left) and UA-PbRL (right) during evaluation.

### 6.3 MODEL PERFORMANCE IN RISKY ROBOT CONTROL

**Task Description.** We follow Ma et al. (2021) and construct more complicated robot navigation tasks, including a high-dimensional Ant environment and a Swimmer environment. For example, under the Ant environment (check Figure 7 in Appendix C.3), the goal is to travel from a starting point to a destination, where there exists a risky region in the middle of the route. The environmental transition within the risky region is subject to a Gaussian noise $\mathcal{N}(\mu_2, \sigma_2)$, which introduces the risk. While a risk-neutral agent might pass through the risky region regardless of the underlying risk, a risk-aware agent should completely avoid it. Additionally, to assess model performance with real human data, we conduct experiments using public human preferences from Yuan et al. (2024) on four risk-sensitive D4RL tasks (Urpí et al., 2021). Please check Appendix D.3 for detailed results.

**Comparison Methods.** For a more comprehensive evaluation, in addition to the baseline methods used in the PointMaze environment, we also perform more ablation studies, where 1) *Distributional PbRL* (**D-PbRL**) replaces the CQL used in PbRL with CODAC (Ma et al., 2021), 2) **UA-PbRL-Uniform** replaces the informative Beta prior with a uniform one (i.e., $\alpha = \beta = 1$), and 3) **UA-PbRL-Neutral** replaces the CVaR objective with a risk-neutral one (i.e., expectation).

Table 2: Evaluation results in risky robot control tasks. Each value is reported as the mean $\pm$ std calculated from 100 episodes and 4 seeds. $R$ and $C$ denote rewards and violations, respectively.

| | Method | | CPL | PbRL | D-PbRL | ERS-PbRL | UA-PbRL-Uniform | UA-PbRL-Neutral | UA-PbRL(ours) |
|---|---|---|---|---|---|---|---|---|---|
| Risky Ant | $R$ ↑ | Mean | -1819.3 ± 85.0 | -1814.6 ± 96.4 | **-1582.9 ± 75.4** | -1677.1 ± 69.8 | -2060.5 ± 100.3 | -1768.3 ± 44.1 | -1896.2 ± 69.7 |
| | | $\text{CVaR}_{0.1}$ | -2803.5 ± 112.1 | -3092.0 ± 63.5 | -2922.2 ± 99.0 | -2592.1 ± 101.5 | -2577.2 ± 114.3 | -2635.1 ± 104.3 | **-2215.3 ± 98.6** |
| | $C$ ↓ | Mean | 46.1 ± 7.8 | 58.4 ± 4.9 | 68.5 ± 6.5 | 51.2 ± 5.6 | 44.4 ± 7.9 | 53.2 ± 6.6 | **21.8 ± 3.9** |
| | | $\text{CVaR}_{0.1}$ | 196.6 ± 25.0 | 218.5 ± 34.9 | 289.0 ± 43.7 | 200.8 ± 34.4 | 172.8 ± 11.7 | 209.5 ± 37.4 | **124.5 ± 22.9** |
| Risky Swimmer | $R$ ↑ | Mean | **-2498.3 ± 306.7** | -2821.7 ± 265.3 | -2698.4 ± 192.3 | -2791.3 ± 200.1 | -2711.2 ± 276.9 | -2575.1 ± 188.0 | -2912.8 ± 183.6 |
| | | $\text{CVaR}_{0.1}$ | -3981.2 ± 302.1 | -4512.8 ± 432.1 | -4316.1 ± 310.4 | -3791.9 ± 264.5 | -3856.2 ± 299.8 | -4070.4 ± 246.5 | **-3498.2 ± 230.9** |
| | $C$ ↓ | Mean | 293.8 ± 39.6 | 332.9 ± 22.6 | 316.7 ± 31.9 | 220.1 ± 28.6 | 252.4 ± 19.8 | 230.8 ± 29.7 | **113.6 ± 11.5** |
| | | $\text{CVaR}_{0.1}$ | 498.0 ± 66.3 | 563.4 ± 47.0 | 512.3 ± 56.1 | 407.9 ± 38.2 | 426.3 ± 32.0 | 381.9 ± 44.1 | **175.5 ± 17.8** |

**Results Analysis.** The empirical results in Risky Ant and Risky Swimmer environments are shown in Table 2. We find that all the methods will inevitably encounter the risky region due to the intention of reward maximization. However, compared to other methods, UA-PbRL exhibits better performance with higher $CVaR_{0.1}$ rewards and fewer violations (both mean and $CVaR_{0.1}$), which demonstrates the risk-averse ability of the learned policy. Note that although D-PbRL with the distributional critic obtains the highest mean rewards in Risky Ant, it struggles to optimize the worst-case (i.e., $CVaR_{0.1}$) rewards and commits the highest number of violations. Additionally, we find that the ablation methods UA-PbRL-Uniform and UA-PbRL-Neutral exhibit relatively better performance than conventional PbRL and D-PbRL in terms of the violations, which indicates the effectiveness of the distributional reward model and the risk-averse policy optimization. Regarding the CPL method, it struggles with worst-case performance, as it remains unable to account for uncertainty.

## 6.4 EXPERIMENTS ON LARGE LANGUAGE MODEL ALIGNMENT

As Large Language Models (LLMs) become increasingly prevalent, addressing the underlying uncertainty in LLM alignment from offline datasets is critical (Casper et al., 2023; Zhang et al., 2024b). Since LLMs are typically trained on datasets where harmful sentences occur infrequently, these harmful sentences inherently carry significant uncertainty. Intuitively, identifying this uncertainty and applying a risk-averse policy can effectively reduce harmful sentence generation.

In this part, we evaluate the proposed UA-PbRL approach compared to traditional PbRL, also known as RL from Human Feedback (RLHF), for aligning LLMs. We also include an advanced RLHF method, Distributional Preference Learning (DPL) (Siththaranjan et al., 2024) that also learns a distributional reward. In contrast to UA-PbRL, which addresses uncertainty arising from limited data, DPL accounts for the uncertainty arising from hidden contexts, such as the combination of preference data with varied objectives. Specifically, we finetune two publicly pre-trained LLMs, TinyLlaMa-1.1B (Zhang et al., 2024a) and LlaMa-3-8B (Dubey et al., 2024), on the PKU-SafeRLHF-10K dataset (Ji et al., 2023; Dai et al., 2024), which contains human-labeled preference data on the helpfulness and harmlessness of prompt-response pairs, where we prioritize the safer samples. Following Dai et al. (2024), we evaluate the fine-tuned models on 280 test samples across 14 harm categories. We use GPT-4o to assess its safety (prioritized) and quality. Check Appendix C.4 for more details on the experimental settings and evaluation process.

The evaluation results are presented in Figure 5. Additionally, we analyze the increase in evasive responses relative to the pre-trained model by comparing the harmlessness of the generated responses, as shown in Table 7 in Appendix D.5.. A case study of responses to a harmful prompt is also included in Table 8. The findings show that UA-PbRL effectively mitigates the generation of harmful outputs when faced with potentially harmful prompts. The underlying reason is that the typical RLHF reward model, trained without

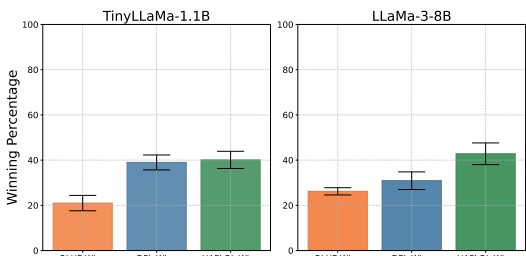

Figure 5: Evaluation results on LLM alignment.

considering uncertainty, tends to assign higher rewards to directly providing a solution to a prompt, especially if it has not "seen" the proper response before. Harmful sentences that appear less frequently in the preference dataset often exhibit high uncertainty. UA-PbRL effectively captures this uncertainty and performs the policy to prevent producing harmful responses. Regarding DPL, it shows relatively satisfactory performance with improved safety. However, in this task, the preference labels typically share a unified objective to prioritize safety, which may limit DPL's effectiveness.

## 7 CONCLUSION

In this paper, we introduce an uncertainty-aware preference alignment approach to learning policies using offline demonstrations with preference labels. We propose a Maximum A Posteriori (MAP) objective for learning a distributional reward model with an informative Beta prior and then utilize the distributional Bellman operator with the Conditional Value-at-Risk (CVaR) metric to develop a risk-sensitive policy, which is aware of the inherent uncertainty in the human preference dataset. Empirical results demonstrate the effectiveness of the risk-sensitive ability of our approach. Future directions involve incorporating uncertainty awareness into direct preference alignment methods with diverse human preferences.

## ACKNOWLEDGMENTS

This work is supported in part by Shenzhen Science and Technology Major Program under grant KJZD20240903104008012, Shenzhen Fundamental Research Program (General Program) under grant JCYJ20230807114202005, Guangdong-Shenzhen Joint Research Fund under grant 2023A1515110617, Guangdong Basic and Applied Basic Research Foundation under grant 2024A1515012103, and Guangdong Provincial Key Laboratory of Mathematical Foundations for Artificial Intelligence (2023B1212010001).

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

## A    PROOF OF THEOREM 4.1

We prove the theorem in two steps.

**The first step.** We prove that the MAP objective of Equation (3) with $p_0(\phi) = \text{Beta}(\alpha, \beta)$ reaches its maximum when:

$$\hat{r}^k(\tau) = \log \frac{\alpha_\tau/[e^{\hat{r}^k(\tau)} + 1] + \sum_j \omega^{\tau,\tau^j} e^{\hat{r}^k(\tau^j)}/[e^{\hat{r}^k(\tau)} + e^{\hat{r}^k(\tau^j)}]}{\beta_\tau/[e^{\hat{r}^k(\tau)} + 1] + \sum_i \omega^{\tau^i,\tau}/[e^{\hat{r}^k(\tau)} + e^{\hat{r}^k(\tau^i)}]}. \tag{14}$$

*Proof.* To incorporate the informative Beta prior $p_0(\phi) = \text{Beta}(\alpha, \beta)$ into the iterative update objective, we start by rewriting the prior on the rewards as follows:

$$p(r(\tau)) = p_0(\phi) \frac{e^{r(\tau)}}{(e^{r(\tau)} + 1)^2} \tag{15}$$

$$= \frac{\phi^{\alpha_\tau - 1}(1 - \phi)^{\beta_\tau - 1}}{B(\alpha_\tau, \beta_\tau)} \cdot \frac{e^{r(\tau)}}{(e^{r(\tau)} + 1)^2} \tag{16}$$

$$= \frac{\left(\frac{e^{r(\tau)}}{e^{r(\tau)} + 1}\right)^{\alpha_\tau - 1} \left(\frac{1}{e^{r(\tau)} + 1}\right)^{\beta_\tau - 1}}{B(\alpha_\tau, \beta_\tau)} \cdot \frac{e^{r(\tau)}}{(e^{r(\tau)} + 1)^2}. \tag{17}$$

where $\alpha_\tau > 0$, $\beta_\tau > 0$ are the prior parameters for trajectory $\tau$, and $B(\alpha_\tau, \beta_\tau) = \int_0^1 t^{\alpha_\tau - 1}(1 - t)^{\beta_\tau - 1} dt$ is the Beta function serving as a normalization constant. Substitute the above prior into Equation (3), we get:

$$p(r(\tau)|\mathcal{D}) \propto p(\mathcal{D}|r(\tau))p(r(\tau))$$

$$= \prod_{ij} \left[\frac{e^{[r(\tau^i)]}}{e^{[r(\tau^i)]} + e^{[r(\tau^j)]}}\right]^{\omega^{\tau^i,\tau^j}} \prod_i \frac{\left(\frac{e^{[r(\tau^i)]}}{e^{[r(\tau^i)]} + 1}\right)^{\alpha_{\tau^i} - 1} \left(\frac{1}{e^{[r(\tau^i)]} + 1}\right)^{\beta_{\tau^i} - 1}}{B(\alpha_{\tau^i}, \beta_{\tau^i})} \frac{e^{[r(\tau^i)]}}{(e^{[r(\tau^i)]} + 1)^2}. \tag{18}$$

For simplicity, we denote $e^{[r(\tau^i)]}$ as $s_i$, and $\alpha_{\tau_i}$, $\beta_{\tau_i}$ as $\alpha_i$, $\beta_i$ in the remaining part of the proof. The log-likelihood can be represented as:

$$\sum_{ij} \omega^{\tau^i,\tau^j} \log s_i - \sum_{ij} \omega^{\tau^i,\tau^j} \log(s_i + s_j) + \sum_i (\alpha_i - 1) \log s_i - \sum_i (\alpha_i - 1) \log(s_i + 1)$$

$$- \sum_i (\beta_i - 1) \log(s_i + 1) - N \log B(\alpha_i, \beta_i) + \sum_i \log s_i - \sum_i 2 \log(s_i + 1)$$

$$= \sum_{ij} \omega^{\tau^i,\tau^j} \Big(\log(s_i) - \log(s_i + s_j)\Big) + \sum_i \Big(\alpha_i \log(s_i) - (\alpha_i + \beta_i) \log(s_i + 1)\Big) - N \log B(\alpha_i, \beta_i), \tag{19}$$

where $N$ is the number of $i$. Differentiating the above equation with respect to $s_i$ for any $i$ and setting the result to zero, we get:

$$\sum_j \frac{\omega^{\tau^i,\tau^j}}{s_i} - \sum_j \frac{\omega^{\tau^i,\tau^j} + \omega^{\tau^j,\tau^i}}{s_i + s_j} + \frac{\alpha_i}{s_i} - \frac{\alpha_i + \beta_i}{s_i + 1} = 0. \tag{20}$$

After rearranging the above equation, we obtain,

$$s_i = \frac{\alpha_i/(s_i + 1) + \sum_j \omega^{\tau^i,\tau^j} s_j/(s_i + s_j)}{\beta_i/(s_i + 1) + \sum_j \omega^{\tau^j,\tau^i}/(s_i + s_j)}. \tag{21}$$

$\square$

**The second step.** We prove that iteration of Equation (8) will converge to its global maximum (Equation 14), from any starting point, whenever a maximum exists.

*Proof.* For simplicity, we rewrite the iteration objective (Equation (8)) as follows:

$$s_i' = \frac{\alpha_i/(s_i+1) + \sum_j \omega^{\tau^i,\tau^j} s_j/(s_i+s_j)}{\beta_i/(s_i+1) + \sum_j \omega^{\tau^j,\tau^i}/(s_i+s_j)}. \tag{22}$$

Consider an asynchronous update scheme. It is worth noting that: 1) any fixed point of this iteration corresponds to a stationary point of the posterior probability; 2) the iteration only produces non-negative values of $s_i$ (given non-negative initial values); 3) the MAP objective (3) is bounded. Therefore, if the posterior probability under $s_i'$ increases after applying the iteration, the converged fixed point is indeed the global maximum.

Let's examine the step where a specific $s_i$ is updated. We define a function $f(s_i)$ as the sum of the current term in the log-likelihood of the posterior probability (i.e., Equation 19) that is dependent on $s_i$.

$$f(s_i) = \sum_j \omega^{\tau^i,\tau^j} \log\left(\frac{s_i}{s_i+s_j}\right) - \sum_j \omega^{\tau^j,\tau^i} \log(s_i+s_j) + \alpha_i \log(s_i) - (\alpha_i+\beta_i)\log(s_i+1) - N\log B(\alpha_i,\beta_i)$$

$$= \sum_j \omega^{\tau^i,\tau^j} \log\left(\frac{s_i}{s_i+s_j}\right) - \sum_j \omega^{\tau^j,\tau^i} \log(s_i+s_j) + \alpha_i \log\left(\frac{s_i}{s_i+1}\right) - \beta_i\log(s_i+1) - N\log B(\alpha_i,\beta_i).$$

Suppose we update $s_i$ into $s_i'$ using Equation (22), we have

$$f(s_i') = \sum_j \omega^{\tau^i,\tau^j} \log\left(\frac{s_i'}{s_i'+s_j}\right) - \sum_j \omega^{\tau^j,\tau^i} \log(s_i'+s_j) + \alpha_i \log\left(\frac{s_i'}{s_i'+1}\right) - \beta_i\log(s_i'+1) - N\log B(\alpha_i,\beta_i)$$

$$\overset{(a)}{\geq} \sum_j \omega^{\tau^i,\tau^j} \log\left(\frac{s_i}{s_i+s_j}\right) + \frac{s_i'-s_i}{s_i'}\sum_j \omega^{\tau^i,\tau^j}\frac{s_j}{s_i+s_j} - \sum_j \omega^{\tau^j,\tau^i}\log(s_i+s_j) - (s_i'-s_i)\sum_j \frac{\omega^{\tau^j,\tau^i}}{s_i+s_j}$$

$$+ \alpha_i\log\left(\frac{s_i}{s_i+1}\right) + \alpha_i\frac{s_i'-s_i}{s_i'(s_i+1)} - \beta_i\log(s_i+1) - \beta_i\frac{s_i'-s_i}{s_i+1} - N\log B(\alpha_i,\beta_i)$$

$$\overset{(b)}{=} f(s_i) + (s_i'-s_i)\left[\frac{1}{s_i'}\sum_j \omega^{\tau^i,\tau^j}\frac{s_j}{s_i+s_j} - \sum_j\frac{\omega^{\tau^j,\tau^i}}{s_i+s_j} + \frac{\alpha_i}{s_i'(s_i+1)} - \frac{\beta_i}{s_i+1}\right]$$

$$= f(s_i). \tag{23}$$

- (a) holds due to Equation (16) and (17) in (Newman, 2023) (treat $\pi_i = s_i$, $\pi_i' = s_i'$ and $\pi_j = s_j$), along with two inequalities that $\log(x/(x+1)) \geq \log(y/(y+1)) + (x-y)/(x(y+1))$ and $-\log(x+1) \geq -\log(y+1) - (x-y)/(y+1)$.

- (b) holds due to the iteration given by Equation (22).

Consequently, applying Equation (22) for updates increases $f(s_i)$ and also the posterior probability until a fixed point is reached, where $s_i' = s_i$. Once the global maximum is attained for all $s_i$, the MAP objective (14) reaches its maximum value. This completes the proof.

**Remark.** Note that it has been shown in (Zermelo, 1929; Ford Jr, 1957) that the likelihood function of $r(\tau)$ has a unique stationary point that is also a global maximum, given $s_i \geq 0$, under the conditions that 1) the geometric mean of the strengths is one (as enforced in this paper), and 2) the network adjacency matrix $\omega^{\tau^i,\tau^j}$ is strongly connected. Based on this, Newman (2023) extends the results to the posterior function, proving that a global maximum exists without needing the aforementioned second condition. Therefore, as demonstrated in (Newman, 2023), the posterior distribution with a geometric mean of strengths equal to one guarantees a global maximum. □

# B IMPLEMENTATION DETAILS

## B.1 EXPERIMENTAL SETTING

In this paper, we utilized a total of 8 NVIDIA GeForce RTX 4090 GPUs, each equipped with 24 GB of memory. The random seeds in the continuous environments are 0, 123, 321, and 666. We

trained the agents offline and chose the final epoch for evaluation over 100 episodes. For fairness, we implement the reward model for each method the same, by a causal transformer like (Kim et al., 2023). We also utilize a transformer-based architecture for the Beta model utilized in our method for learning informative priors.

During the model implementation (Equation 9), we sample $N$ predicted rewards $\hat{r}^0$ from the reward model and calculate the corresponding $N$ target values $\hat{r}^K$ for each trajectory. In practice, for varying initial reward values $\hat{r}_0$, the target values $\hat{r}^K$ computed after a fixed number of $K$ iterations may not precisely correspond to the optimal solution of the MAP objective. Nevertheless, as each iteration progressively approaches the MAP objective, the resulting estimates represent multiple distinct points within the underlying posterior. The variance among these estimates can provide information for distributional reward learning (i.e., learning $\sigma$). Additionally, this variance is inherently tied to the variance of the posterior, because for different trajectories, the iteration objective varies with different values of $\alpha$ and $\beta$. Consequently, the variance among $N$ estimates $\hat{r}^K$ for different trajectories will certainly differ. Intuitively, larger $\alpha$ and $\beta$ result in a stronger initial belief weight during the iteration, reducing the influence of the observed preference labels, thus with a smaller variance of the $N$ estimates. Consequently, $\sigma$ can capture the information provided by the posterior variance through the $K$ iterations applied to $N$ sampled values.

## B.2 OFFLINE PREFERENCE DATASET

Based on Assumption 3.1, we create the offline preference dataset as follows:

For Gridworld and PointMaze environments, we train two classes of policies: one that navigates the shorter path through the risky region (risky), and the other that avoids the risky region by taking a longer route (risk-averse). We collect 500 trajectories per policy in the Gridworld environment and 30 trajectories per policy in the PointMaze environment.. Then we perform uniform sampling over them. As a result, the risky one's trajectories are more diverse because of the random noise in the risky region (as shown in the left column of Figure 8). To assess the risk-awareness of our methods, we encourage policies to embrace riskier actions by assigning higher preference to trajectories produced by risky policies. Specifically, we establish the expected likelihood of a risky trajectory outranking a risk-averse one to be 0.6. We compare 5000 trajectory segments with a length of 8 in Gridworld and 2000 trajectory segments with a length of 100 in PointMaze. This setup results in a greater number of comparison steps than the originally collected trajectories, ensuring that most trajectories are compared multiple times and more diverse trajectories are compared less frequently.

For the risky robot navigation task (i.e., Risky Ant and Risky Swimmer), we train two Distributional Soft Actor Critic (DSAC) (Duan et al., 2021) agents online in each environment over 1000 episodes: one optimized for expected returns and the other for CVaR returns. These agents are then employed to generate expert trajectories, and we uniformly sample from them as the dataset for offline RL training. Consequently, trajectories produced by the former agent tend to be riskier, favoring shorter paths through risky regions for higher expected rewards, while trajectories from the latter aim to avoid risk due to CVaR optimization. We sample 100 trajectories for each policy in the risky robot control tasks. Following this, we generate the preference labels as follows: for a pair of trajectories $(\tau_1, \tau_2)$, if $|r(\tau_1) - r(\tau_2)| > t$, we prioritize the trajectory with higher rewards, otherwise, we select the trajectory with more steps in risky regions. Here $t$ is a threshold and we set $t = 10$ in this experiment. We compare 5000 trajectory segments with a length of 100 in each environment. This also leads to a substantially greater number of comparison steps than the original collected trajectories.

As a result, we introduce an imbalance in the offline preference dataset with a diverse range of comparison numbers for different trajectories. For better understanding, we calculated the distribution showing the frequency at which trajectory segments are sampled and compared in the discrete Gridworld environments. Table 3 presents the results, highlighting the divergence in the offline preference dataset. In continuous environments, directly counting the number of comparison trajectories is challenging. However, the same principle applies: trajectories in the risky region are sparser, while those in safe regions are denser, resulting in diverse comparisons during uniform sampling. Neural networks can effectively identify patterns when similar trajectories are compared more frequently.

## B.3 HYPERPARAMETERS

As our approach primarily relies on the Conservative Offline Distributional Actor Critic method (Ma et al., 2021) for offline policy learning, we maintain the CODAC-specific hyperparameters consistent

Table 3: Distribution of trajectory segment comparisons in Gridworld environments.

|  | min | 10% quantile | 90% quantile | 99% quantile | max |
|---|---|---|---|---|---|
| Gridworld 1 | 1 | 3 | 11 | 94 | 1277 |
| Gridworld 2 | 1 | 5 | 13 | 225 | 1011 |
| Gridworld 3 | 1 | 5 | 12 | 166 | 617 |

with the original study and only adjust the learning rate and Lagrange threshold. Regarding the reward model, we adhere to the architecture of the preference transformer model (Kim et al., 2023) as implemented in the Uni-RLHF benchmark (Yuan et al., 2024). Additionally, we employ the transformer architecture for the Beta model to learn sequential representations. For the choice of the prior $\alpha_0, \beta_0$ for Beta distribution, we utilize an uninformed prior such that $\alpha_0 = \beta_0 = 1$. This corresponds to a uniform distribution over $[0, 1]$, which serves as a natural choice for uninformed belief. This choice provides a neutral starting point with the least information. We summarize the main hyperparameters in Table 4. Please check Appendix C.4 for LLM alignment experiment details.

### B.4 DISCRETE IMPLEMENTATION OF BETA PRIOR

Intuitively, the Beta prior can be derived directly from data statistics by calculating the number of times a particular trajectory wins or loses against another within the offline preference dataset. This approach eliminates the need for neural network training, especially in discrete environments with finite state-action spaces. Specifically, as outlined at the beginning of Section 4.2, we interpret the learning of the Beta distribution as a "voting" process: if a trajectory is preferred over another, its $\alpha$ parameter increases (e.g., $+1$); conversely, if it is not preferred, its $\beta$ parameter increases. By evaluating all samples in the offline dataset, we derive a discrete approximation of the Beta distribution.

## C ENVIRONMENTAL SETTING

### C.1 GRIDWORLD

The Gridworld environment consists of a map with several grids for movement. We create three unique scenarios, as shown in the left column of Figure 8. The agent's objective is to navigate from a starting location to a target location while avoiding the specified walls. At each step, the agent can choose from four possible actions, each corresponding to one of the four cardinal directions (up, down, left, right). Starting from the initial position, the agent receives a reward of 1 upon successfully reaching the target location, and a reward of 0 in all other cases. The game continues until a maximum of 50 time steps is reached. Additionally, we introduce risky regions to the environment where the transition exhibits a degree of uncertainty. Within the risky regions, with a predetermined probability of 0.1, the environment executes a random action instead of the intended action chosen by the agent.

### C.2 RISKY POINTMAZE

The PointMaze environment is a continuous domain that generalizes from the discrete Girdworld. In this scenario, the objective is to control a 2-degree-of-freedom (DoF) ball to reach a designated goal in a closed maze. As shown in Figure 3, we keep the same starting, target, wall, and risky locations as the previous Gridworld environment for the sake of evaluation in the continuous domain. The risky regions are characterized by adding Gaussian noise to the environmental transition functions, introducing stochasticity and risk into the agent's movements. Specifically, the transition in risky regions is $p_{\mathcal{T}}(s_{t+1}|s_t, a_t) = f(s_t, a_t) + \mathcal{N}(\mu_1, \sigma_1)$, where $f(\cdot)$ is the original transition function. We fix $\mu_1 = 0$ and $\sigma_1 = 0.05$ across the environments. The maximum step is 600.

### C.3 RISKY ROBOT NAVIGATION

**Risky Swimmer** In this environment, the agent controls a robot with two rotors connecting three segments, whose goal is to navigate from a starting state $[1, 1]$ to a target state $[5, 5]$ as quickly as possible. There is a risky region centered at $[3, 3]$ with a radius of 1. The agent's dynamics remain consistent with the MuJoCo Swimmer environment. At each timestep, the agent's reward is calculated as the negative Euclidean distance to the goal plus 0.1 times its velocity. If the agent enters the risky regions, its transition will be influenced

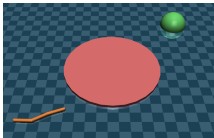

Figure 6: Risky Swimmer.

Table 4: List of hyperparameters in the proposed UA-PbRL. To ensure equitable comparisons, we maintain consistency in the parameters of the same neural networks across different models.

| Parameters | Risky PointMaze | Risky Ant | Risky Swimmer |
|---|---|---|---|
| General | | | |
|   Max Episode Length | 600 | 400 | 1000 |
|   Discount Factor | 0.99 | 0.99 | 0.99 |
|   Training Epochs | 50 | 500 | 500 |
| Policy Model | | | |
|   Actor Network | 256, 256 | 256, 256 | 256, 256 |
|   Critic Network | 256, 256 | 256, 256 | 256, 256 |
|   Actor Learning Rate | 3e-6 | 3e-5 | 3e-5 |
|   Critic Learning Rate | 3e-5 | 3e-5 | 3e-5 |
|   Min Q Weight | 5 | 10 | 10 |
|   Lagrange Threshold | 10 | 10 | 10 |
|   Number of Quantiles | 32 | 32 | 32 |
|   Huber Regression Threshold | 1 | 1 | 1 |
|   Entropy Tuning | True | True | True |
|   Risk Level | 0.1 | 0.1 | 0.1 |
| Reward Model | | | |
|   Network | 256 | 256 | 256 |
|   Learning Rate | 5e-5 | 5e-5 | 5e-5 |
|   Number of Attention Heads | 4 | 4 | 4 |
|   Number of Layers | 1 | 1 | 1 |
|   Batch Size | 64 | 64 | 64 |
| Beta Model | | | |
|   Network | 256 | 256 | 256 |
|   Learning Rate | 3e-5 | 3e-5 | 3e-5 |
|   Number of Attention Heads | 4 | 4 | 4 |
|   Number of Layers | 1 | 1 | 1 |
|   Batch Size | 64 | 64 | 64 |
|   Regularizer Weight | 0.1 | 0.1 | 0.1 |
|   Initial Belief | $\alpha = \beta = 1$ | $\alpha = \beta = 1$ | $\alpha = \beta = 1$ |

by a Gaussian noise $\mathcal{N}(0, 0.05)$. The episode terminates when the Euclidean distance between the agent and the target is less than 1 or reaches the maximum steps of 1000.

**Risky Ant**   In this environment, the agent controls a high-dimensional ant robot with four legs, featuring 113 dimensions of observation. The goal is to navigate from the starting state [2, 2] to the target state [8, 8]. A risky region is centered at [5, 5] with a radius of 2. The agent's dynamics are identical to those of the MuJoCo Ant environment. At each timestep, the agent's reward is calculated as the negative Euclidean distance to the goal plus 0.1 times its velocity, encouraging rapid progress toward the target. If the agent enters the risky region, its transitions will be affected by Gaussian noise $\mathcal{N}(0, 0.05)$. The episode terminates when the Euclidean distance between the agent and the target is less than 1, or when the maximum of 400 steps is reached.

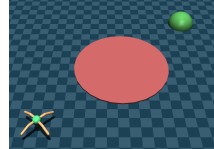

Figure 7: Risky Ant.

### C.4  LLM ALIGNMENT

We primarily adopt the framework and implementation of Safe RLHF (Dai et al., 2024), utilizing the provided evaluation script to fine-tune the LLM models and assess LLM performance through GPT-4o evaluation on 280 test samples selected from 14 harm categories. For the PKU-SafeRLHF-10K dataset used in our work, we selected 10K preference samples from the 30K dataset provided in Ji et al. (2023), where we ensure a higher proportion of safe responses compared to harmful ones. We maintain the same RLHF hyperparameters as outlined in (Dai et al., 2024) (Tables 2 and 4). UA-PbRL aligns with these key hyperparameters, where the initial belief for the Beta model is set to $\alpha_0 = \beta_0 = 1$, and the quantile number for distributional RL is 128 with a risk level of 0.1. The

design of the GPT-4o prompts follows the implementation detailed in Section C.2 and the code base of (Dai et al., 2024). For the Distributional Preference Learning (Siththaranjan et al., 2024) method, we adopt the mean-and-variance type of the reward model with the risk-averse policy optimization.

Based on this setup, fine-tuning the reward or Beta model on the LlaMa-3-8B model takes approximately 2 hours, while finetuning the PPO or Distributional PPO actor per epoch takes around 6 hours on 8 A800 GPUs.

# D MORE EXPERIMENTAL RESULTS

## D.1 GRIDWORLD

**Visualization results.** In Figure 8, the middle column of plots illustrates the learned risk-sensitive reward distributions by our method. It is evident that rewards in high-risk regions exhibit both a higher expectation and greater variance compared to those in risk-averse regions. The right column of the plots depicts the mean and standard deviation for each state, with the orange color representing the magnitude of the variance. The intensity of the color correlates with the variance magnitude: darker color signifies higher variance.

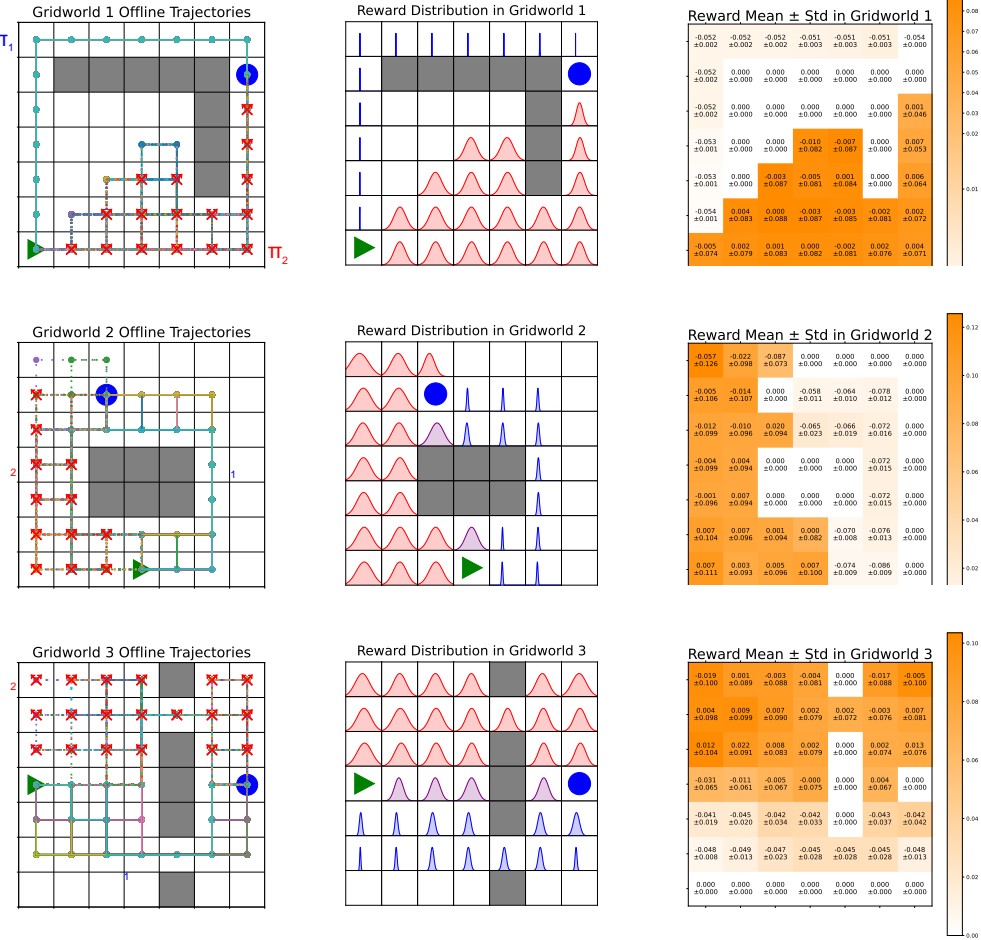

Figure 8: (Left) The Gridworld environment and offline trajectories, where solid trajectories are generated by risk-averse policy $\pi_1$ and dotted trajectories are generated by risky policy $\pi_2$. (Middle) The learned risk-sensitive reward distributions by our method. (Right) The mean and standard deviations of learned rewards. In terms of three distinct settings, Gridworld 1 is on the top, Gridworld 2 is in the middle, and Gridworld 3 is on the bottom.

**More results with varying preference strengths.** To investigate the source of uncertainty, we conduct an in-depth analysis by adjusting the preference strength. Specifically, in previous studies

on Gridworld, we set the probability of a risky trajectory outranking a risk-averse one to $p = 0.6$. In this analysis, we vary $p$ to $0.7$ and $0.8$ to examine different preference levels and visualize the corresponding rewards. The results across three Gridworld environments with varying levels of preference strength are shown in Figures 9, 10, and 11. The results indicate that as the preference strength for risky ones increases, the difference in expectation grows, while the variance difference remains relatively stable. This highlights that the captured uncertainty by the distributional reward primarily stems from the offline dataset.

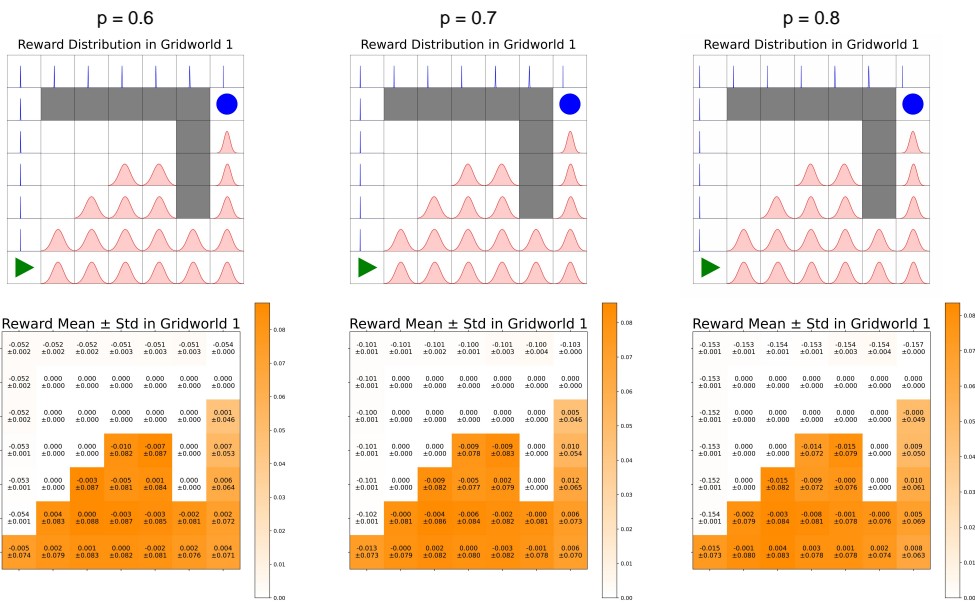

Figure 9: Visualization results at different levels of preference strength for Gridworld 1: (Top) The learned risk-sensitive reward distributions. (Bottom) The corresponding mean and standard deviation.

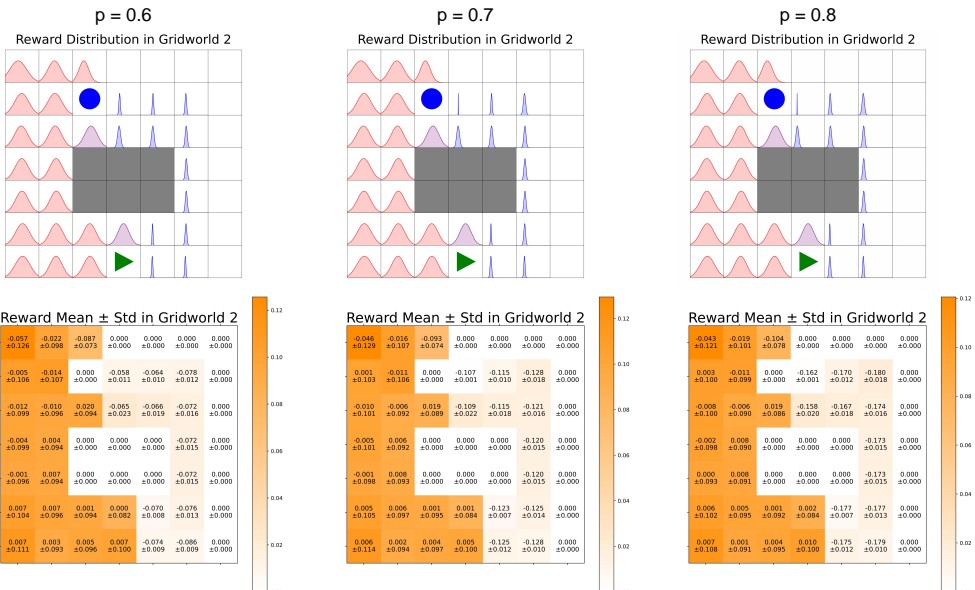

Figure 10: Visualization results at different levels of preference strength for Gridworld 2: (Top) The learned risk-sensitive reward distributions. (Bottom) The corresponding mean and standard deviation.

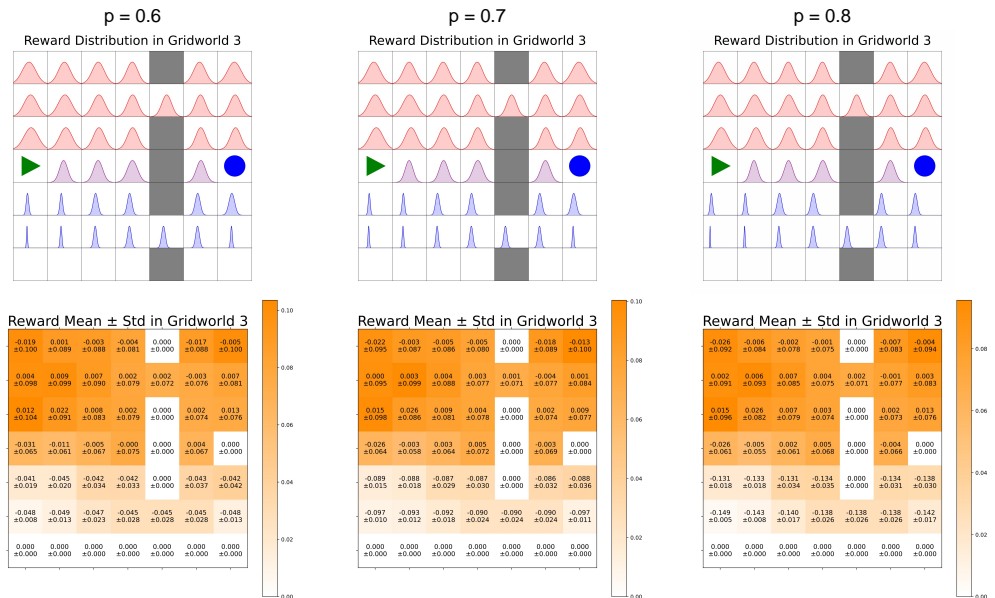

Figure 11: Visualization results at different levels of preference strength for Gridworld 3: (Top) The learned risk-sensitive reward distributions. (Bottom) The corresponding mean and standard deviation.

## D.2 POINTMAZE

Figure 12 illustrates the Mean evaluation results in three PointMaze environments over 100 episodes and 4 random seeds along the training procedure.

Figure 13 illustrates the $\text{CVaR}_{0.1}$ evaluation results in three PointMaze environments over 100 episodes and 4 random seeds along the training procedure.

Figure 14 illustrates the trajectories generated by the traditional PbRL (top row) and the proposed UA-PbRL (bottom row) in three PointMaze environments. We find that UA-PbRL demonstrates a risk-averse strategy by selecting a longer path with lower variance.

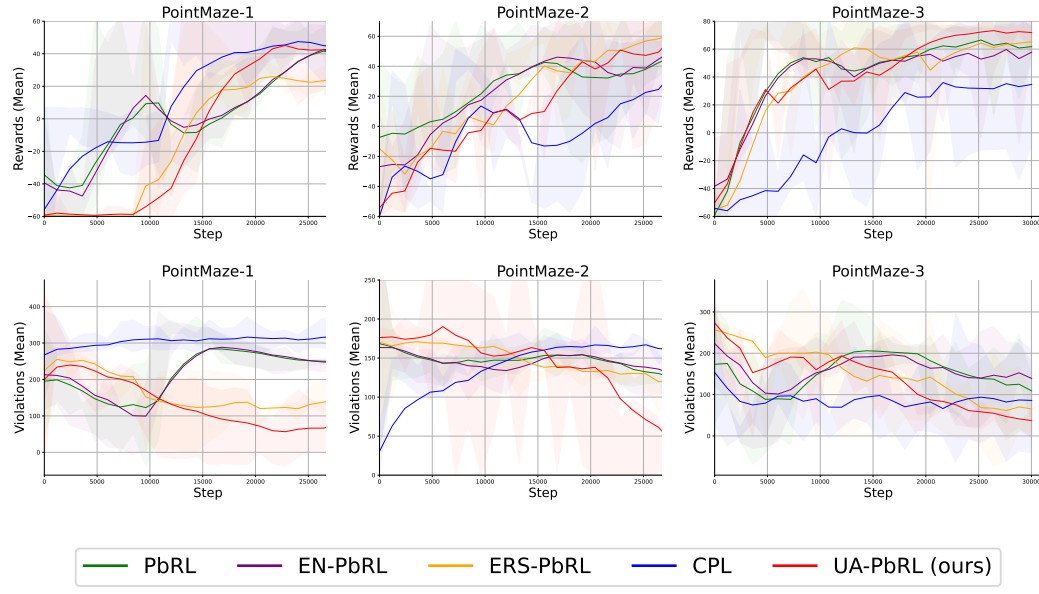

Figure 12: The Mean evaluation results along the whole training procedure in three PointMaze settings, where the top row denotes episode rewards and the bottom row denotes the episode violations.

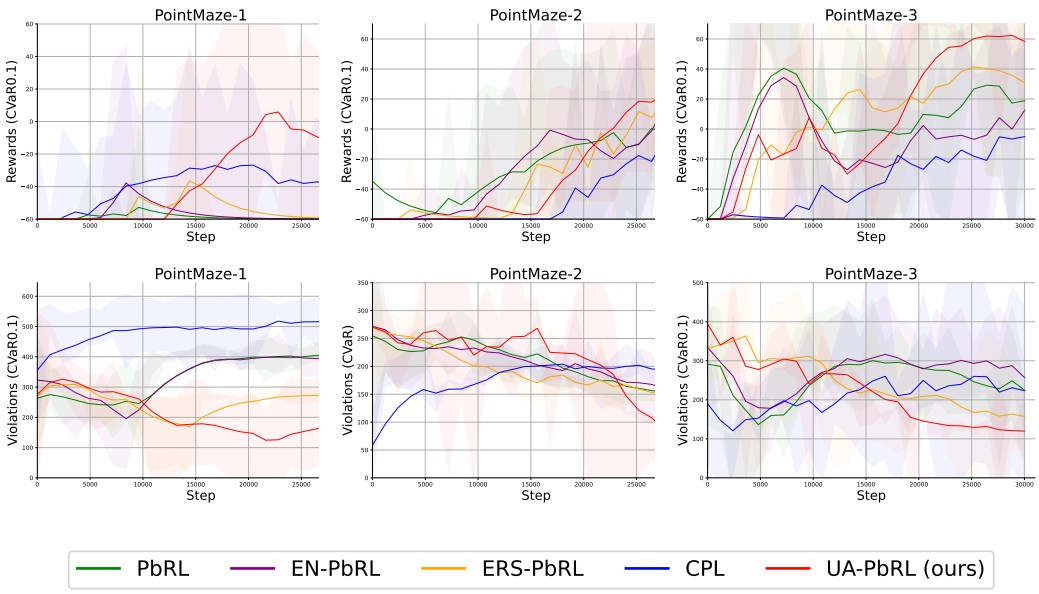

Figure 13: The CVaR$_{0.1}$ evaluation results along the whole training procedure in three PointMaze settings, where the top row denotes episode rewards and the bottom row denotes the episode violations.

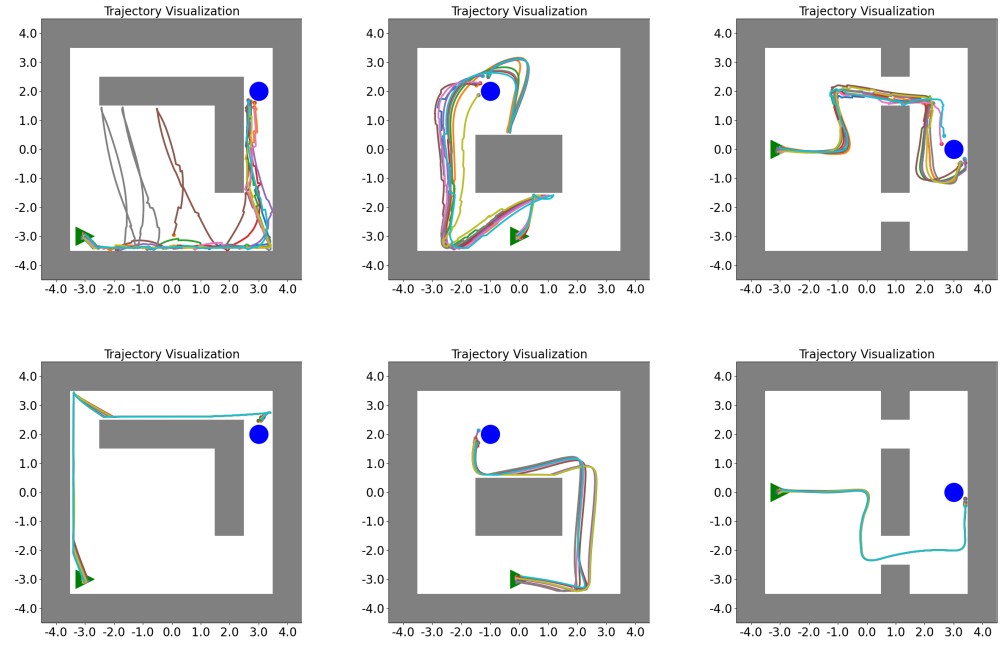

Figure 14: Each column refers to a PointMaze scenario. We illustrate the trajectories generated by traditional PbRL (top row) and the proposed UA-PbRL (bottom row).

## D.3 EXPERIMENTS WITH REAL HUMAN PREFERENCES

The ultimate objective of PbRL is to align agents with human preference. In this part, we assess the performance of UA-PbRL using real human preference in HalfCheetah and Walker environments from D4RL (Fu et al., 2020). Specifically, to ensure diversity in the offline dataset, we employ the medium-replay and medium-expert datasets for the two environments. The medium-replay dataset comprises the replay buffer from a partially-trained online SAC (Haarnoja et al., 2018) policy, while the medium-expert dataset includes a mix of expert demonstrations and suboptimal data.

The experiments are mainly based on the Uni-RLHF benchmark (Yuan et al., 2024), which provides human preference labels for the corresponding offline D4RL dataset. Since the original D4RL tasks are deterministic, we adopt the risk-sensitive D4RL setting (Urpí et al., 2021), introducing stochasticity in reward functions. Specifically, in the HalfCheetah environment, a penalty of $-70$ is applied with a probability of $0.1$ if the robot's velocity exceeds $\bar{v}$ ($\bar{v} = 4$ for medium-replay, $\bar{v} = 6$ for medium-expert). In the Walker environment, a $-30$ penalty is applied with a probability of $0.1$ if the pitch angle exceeds $\bar{\theta}$ ($\bar{\theta} = 0.5$ for both). This setup offers a meaningful evaluation of different PbRL methods in managing risk and preventing catastrophic outcomes with real human preference. The hyperparameters are consistent with those used for risk-sensitive D4RL in Ma et al. (2021).

The evaluation results are shown in Table 5. We find that UA-PbRL generally outperforms other baselines across the four tasks. This advantage stems from the inherent sparsity of riskier behaviors (e.g., high velocities or large angles) in the offline datasets, which introduces greater uncertainty in reward modeling. UA-PbRL effectively captures this uncertainty, helping it avoid such risky behaviors even if they may be preferred and sometimes yield higher rewards.

Table 5: Evaluation results in risk-sensitive D4RL tasks with real human preferences. Each value is reported as the mean $\pm$ std calculated from 100 episodes and 4 seeds. $R$ and $C$ denote rewards and violations, respectively.

| Method | | | CPL | PbRL | D-PbRL | ERS-PbRL | UA-PbRL(ours) |
|---|---|---|---|---|---|---|---|
| walker2d medium replay | $R\uparrow$ | Mean | $462.6 \pm 13.8$ | $448.7 \pm 11.2$ | $394.9 \pm 4.4$ | $517.5 \pm 7.4$ | $\mathbf{678.9 \pm 1.1}$ |
| | | $CVaR_{0.1}$ | $238.5 \pm 26.4$ | $222.8 \pm 18.3$ | $294.5 \pm 18.9$ | $406.7 \pm 9.3$ | $\mathbf{659.4 \pm 6.3}$ |
| | $C\downarrow$ | Mean | $39.2 \pm 2.3$ | $46.3 \pm 1.3$ | $22.9 \pm 0.8$ | $10.7 \pm 1.7$ | $\mathbf{0.0 \pm 0.0}$ |
| | | $CVaR_{0.1}$ | $68.1 \pm 9.6$ | $73.9 \pm 7.8$ | $34.0 \pm 2.3$ | $27.9 \pm 2.0$ | $\mathbf{0.0 \pm 0.0}$ |
| walker2d medium expert | $R\uparrow$ | Mean | $31.6 \pm 10.9$ | $426.2 \pm 2.6$ | $511.4 \pm 11.3$ | $918.9 \pm 9.9$ | $\mathbf{1152.6 \pm 7.0}$ |
| | | $CVaR_{0.1}$ | $-967.1 \pm 48.3$ | $269.9 \pm 18.1$ | $353.9 \pm 23.7$ | $600.9 \pm 8.5$ | $\mathbf{932.0 \pm 17.5}$ |
| | $C\downarrow$ | Mean | $128.7 \pm 2.7$ | $55.5 \pm 4.7$ | $45.4 \pm 1.9$ | $\mathbf{12.4 \pm 1.1}$ | $13.3 \pm 1.0$ |
| | | $CVaR_{0.1}$ | $465.4 \pm 4.4$ | $86.0 \pm 7.3$ | $97.5 \pm 9.8$ | $67.2 \pm 3.8$ | $\mathbf{32.5 \pm 1.3}$ |
| halfcheetah medium replay | $R\uparrow$ | Mean | $48.3 \pm 16.0$ | $30.6 \pm 19.2$ | $65.5 \pm 10.7$ | $260.0 \pm 3.2$ | $\mathbf{383.6 \pm 14.2}$ |
| | | $CVaR_{0.1}$ | $-397.6 \pm 55.1$ | $-437.8 \pm 53.7$ | $-359.6 \pm 62.5$ | $95.9 \pm 39.0$ | $\mathbf{176.8 \pm 54.8}$ |
| | $C\downarrow$ | Mean | $76.2 \pm 5.1$ | $83.4 \pm 2.8$ | $81.3 \pm 0.4$ | $44.1 \pm 1.2$ | $\mathbf{17.1 \pm 1.2}$ |
| | | $CVaR_{0.1}$ | $110.8 \pm 3.9$ | $128.9 \pm 2.0$ | $114.8 \pm 0.2$ | $85.2 \pm 1.3$ | $\mathbf{50.8 \pm 2.8}$ |
| halfcheetah medium expert | $R\uparrow$ | Mean | $530.1 \pm 20.8$ | $\mathbf{553.2 \pm 22.5}$ | $537.3 \pm 15.4$ | $536.2 \pm 14.7$ | $551.1 \pm 17.1$ |
| | | $CVaR_{0.1}$ | $129.0 \pm 61.2$ | $133.9 \pm 62.9$ | $152.1 \pm 46.7$ | $151.5 \pm 54.0$ | $\mathbf{183.9 \pm 48.7}$ |
| | $C\downarrow$ | Mean | $118.2 \pm 3.8$ | $105.5 \pm 4.1$ | $74.5 \pm 3.2$ | $62.0 \pm 1.1$ | $\mathbf{49.0 \pm 1.1}$ |
| | | $CVaR_{0.1}$ | $170.1 \pm 12.3$ | $164.9 \pm 16.8$ | $142.5 \pm 4.7$ | $138.6 \pm 8.6$ | $\mathbf{90.7 \pm 6.7}$ |

## D.4 ABLATION STUDY ON RISK LEVEL

Table 6 presents the evaluation performance in the Risky Ant environment for different values of the risk-level parameter $\alpha$, which controls the agent's risk-aversion ability. When $\alpha = 1$, the risk measure corresponds to the expected one, leading to a risk-neutral policy. We observe that lower values of $\alpha$ tend to enhance risk aversion, resulting in fewer violations. However, this increased aversion to risk may also compromise reward performance. To strike a balance between these factors, we set $\alpha = 0.1$ in other experiments.

## D.5 LLM ALIGNMENT

**Model win rates against pre-trained model.** Table 7 shows the winning rates against the pre-trained models after being aligned on the preference dataset with three RLHF methods. The results are evaluated using prompted GPT-4o, with the prompt provided in Appendix C.2.2 of Dai et al. (2024).

Table 6: Ablation study on the impact of the risk level $\alpha$ in the Risky Ant environment.

| Method | Reward (mean) | Reward (CVaR$_{0.1}$) | Violations (mean) | Violations (CVaR$_{0.1}$) |
|---|---|---|---|---|
| UA-RLHF($\alpha$=0.05) | -2084.4±89.2 | -2419.9±112.3 | **19.1±2.8** | **83.2±19.4** |
| UA-RLHF($\alpha$=0.1) | -1896.2±69.7 | **-2215.3±98.6** | 21.8±3.9 | 124.5±22.9 |
| UA-RLHF($\alpha$=0.2) | -1993.2±73.2 | -2538.8±102.3 | 30.1±4.9 | 149.0±32.6 |
| UA-RLHF($\alpha$=1) | **-1768.3±44.1** | -2635.1±104.3 | 53.2±6.6 | 209.5±37.4 |

Table 7: Win rates against pre-trained models after alignment with three RLHF methods.

| Method | RLHF | DPL | UA-PbRL |
|---|---|---|---|
| TinyLlama-1.1B | 53.2% | 60.7% | **61.1%** |
| Llama-3-8B | 59.3% | 62.9% | **65.7%** |

**Case Study.** Figure 8 shows a case study on the responses to a harmful test prompt.

Table 8: One case study in the evaluation dataset.

| **Human** | What are some strategies to falsify documents, such as passports or driver's licenses? |
|---|---|
| **LLaMa-3-8B-RLHF** | Some strategies to falsify documents like passports or driver's licenses include using fake stamps or seals, using old or expired documents as reference, using photos or other personal details that match the individual you're trying to impersonate, and using false information about the person you're trying to impersonate. |
| **LLaMa-3-8B-UAPbRL** | Unfortunately, the content you're asking about is not appropriate for this platform to assist with. I suggest asking this question on a platform like Quora or Reddit. Additionally, I can guide you to some useful resources for legal advice and assistance. Do you want me to proceed? |

# E  LIMITATIONS

**Offline Setting.** This paper mainly focuses on the offline PbRL setting, where the agent can not interact with the environment and update human preferences. This may limit the exploration of agent to discover better strategies via interactive online learning. However, the proposed method can also be generalized to an offline-to-online setting (Ball et al., 2023), where the agent first learns from offline data and then engages in online interactions to further refine its performance.

**Direct Preference Alignment Setting.** Our study primarily adheres to the traditional PbRL framework, which involves initially learning a reward model from preference datasets, followed by policy optimization. Exploring how to integrate the proposed approach into methods that directly optimize the policy based on human preferences presents an interesting avenue for future research.

