# OpenReview forum: "A Distributional Approach to Uncertainty-Aware Preference Alignment Using Offline Demonstrations"
_ICLR.cc/2025/Conference — ICLR 2025 Poster_

### Official Review · Reviewer_oXZj · 2024-11-03

**Soundness:** 2
**Presentation:** 3
**Contribution:** 3
**Rating:** 6
**Confidence:** 3

**Summary:**

This paper introduces UA-PbRL, which employs a distributional reward model to capture uncertainty in offline PbRL settings. This distributional model is then leveraged to train a risk-sensitive policy. The paper presents a MAP objective for reward learning, incorporating Beta priors to account for uncertainty, which is learned via variational inference. Additionally, it utilizes a distributional Bellman operator and the CVaR metric for policy optimization. Empirical evaluations across various tasks demonstrate the effectiveness of UA-PbRL compared to other PbRL baselines.

**Strengths:**

The paper is well-organized and addresses a significant problem in PbRL. The proposed method is evaluated across a variety of tasks, demonstrating its effectiveness.

**Weaknesses:**

Please see Questions.

**Questions:**

1. In line 71, preference assignment -> preference alignment.

2. Clarification on Problem Setting and Example: Is the uncertainty mainly due to too few comparisons rather than imbalance in the dataset? This remains unclear to me. Could you provide further clarification?

3. In the motivation example for the imbalanced preference dataset, what do you mean by "outlier trajectories"? Based on Assumption 3.1, all $N_1+N_2$ trajectories should have an equal probability of being selected to form a pair. The authors state that "a specific trajectory such as $\tau_2^\prime$ and other outlier trajectories are less likely to be selected during sampling from the dataset." Could you clarify why this is the case?

4. The derivation of Equation 2, particularly the first step, is unclear to me. Since this is a crucial part of the approach, please provide more detail in the main text and the appendix if necessary.

5. In Section 4.2, please clarify what is meant by "positive" and "negative" human feedback.

6. It seems that the authors aim to learn a data-driven prior distribution of $\phi$ using a variational distribution, with the marginal distribution of $\phi$ defined by $\alpha_0, \beta_0$. Do varying choices of $\alpha_0$ and $\beta_0$ significantly impact prior learning and subsequent reward and policy learning?

7. Is variational inference computationally expensive, and does it risk converging to a local optimum?

8. The authors refer to two prior works on PbRL that especially focus on confidence and uncertainty [1,2], which could serve as reasonable baselines. Why were these approaches not included for comparison?

9. The uncertainty of preference alignment in LLMs may have different motivations compared to control tasks. A brief discussion on this distinction in the introduction or experiment section would be beneficial.

10. The paper lacks an evaluation on offline dataset collected from real human feedback.


[1] Wanqi Xue, Bo An, Shuicheng Yan, and Zhongwen Xu. Reinforcement learning from diverse human preferences. arXiv preprint arXiv:2301.11774, 2023.

[2] Jie Cheng, Gang Xiong, Xingyuan Dai, Qinghai Miao, Yisheng Lv, and Fei-YueWang. Rime: Robust preference-based reinforcement learning with noisy preferences. arXiv preprint arXiv:2402.17257, 2024.

---

> ### Author Response · Authors · 2024-11-21
> **Author response to Reviewer oXZj - Part 1**
>
> Dear Reviewer, we sincerely appreciate your constructive feedback and thank you for recognizing the significance of our work. We have carefully considered your suggestions, and we hope that the following response can address your concerns:
>
> > Q1. In line 71, preference assignment $\rightarrow$ preference alignment.
>
> **A1.** Thanks for pointing out. We have corrected it in the revised paper.
>
> ---
>
> > Q2. Clarification on Problem Setting and Example: Is the uncertainty mainly due to too few comparisons rather than imbalance in the dataset? This remains unclear to me. Could you provide further clarification?
>
> **A2.** Thank you for this thoughtful question. To clarify, given an offline dataset of fixed size, **imbalance indicates some trajectories will have relatively fewer comparisons than others.** Specifically, as stated in Assumption 3.1, the imbalance or diversity within the trajectory dataset results in varying numbers of comparisons for different trajectory pairs during the uniform sampling process. More diverse trajectories will undergo fewer human comparisons, leading to greater uncertainty and lower confidence in their evaluation.
>
> ---
>
> > Q3. In the motivation example for the imbalanced preference dataset, what do you mean by "outlier trajectories"? Based on Assumption 3.1, all $N_1+N_2$ trajectories should have an equal probability of being selected to form a pair. The authors state that "a specific trajectory such as $\tau^{\prime}_{2}$ and other outlier trajectories are less likely to be selected during sampling from the dataset." Could you clarify why this is the case?
>
> **A3.** Thank you for pointing this out. When we refer to "outlier trajectories," we indicate the trajectories with **small densities in the data distribution**. The small density is due to their diverse behavior compared to others in the dataset. These "outlier trajectories" may arise from inherent stochasticity in the generating policy or the dynamics of the environment.
>
> Regarding Assumption 3.1, you are correct that all $N_1+N_2$ trajectories have an equal probability of being selected to form a pair. However, while the selection probability is indeed uniform, these "outlier trajectories" are less likely to appear frequently in the comparison process due to their greater diversity. As a result, they are selected less often for comparisons with other trajectories.
>
> For instance, in Figure 1, consider comparing each pair of the 3 sample trajectories generated by $ \pi_1 $ and $ \pi_2 $. Since $ \tau_1, \tau_1^\prime, \tau_1^{\prime\prime}$ are quite similar (in this case, $ \tau_1=\tau_1^\prime= \tau_1^{\prime\prime}$), while $ \tau_2, \tau_2^\prime, \tau_2^{\prime\prime} $ are more diverse (so that $ \tau_2\neq \tau_2^\prime\neq\tau_2^{\prime\prime} $), the resulting comparisons would show that $ \tau_1$) is compared 3 times, whereas $ \tau_2, \tau_2^\prime, \tau_2^{\prime\prime} $ are each compared only once.
>
> ---
>
> > Q4. The derivation of Equation 2, particularly the first step, is unclear to me. Since this is a crucial part of the approach, please provide more detail in the main text and the appendix if necessary.
>
> **A4.** Sorry for the misunderstanding. In Equation 2, the notation $d$ refers to the derivative operation. And as defined in Lines 224-225, $\phi(\tau) = \frac{e^{r_\varphi(\tau)}}{e^{r_\varphi(\tau)} + 1}$ (we use $\phi$ as shorthand). By applying the chain rule, we can derive the first two equations, and the last equation can be derived with deviation.
>
> ---
>
> > Q5. In Section 4.2, please clarify what is meant by "positive" and "negative" human feedback.
>
> **A5.** Thank you for your question. In Section 4.2, "positive" human feedback refers to cases where one trajectory is preferred over another, while "negative" feedback indicates that one trajectory is ranked lower. We have clarified these definitions in the revised manuscript.

---

> ### Author Response · Authors · 2024-11-21
> **Author response to Reviewer oXZj - Part 2**
>
> > Q6. It seems that the authors aim to learn a data-driven prior distribution of $\phi$ using a variational distribution, with the marginal distribution of $\phi$ defined by $\alpha_0,\beta_0$. Do varying choices of $\alpha_0,\beta_0$ significantly impact prior learning and subsequent reward and policy learning?
>
> **A6.** Thank you for raising this question. We agree that developing a mechanism to study the choices of the prior $\alpha_0,\beta_0$ is critical. In this work, we utilize an **uninformed prior such that $\alpha_0 = \beta_0 = 1$.** This corresponds to a uniform distribution over \([0, 1]\), which serves as a natural choice for uninformed belief. This choice provides **a neutral starting point with the least information.**
>
> In terms of varying $\alpha_0,\beta_0$ to other values, this operation indeed has an impact on the reward modeling. For instance, setting $\alpha_0 = 99999$ and $\beta_0 = 1$ will introduce strong prior information into the model (implying 99999 persons vote 1, and only one person votes 0). Therefore, setting $\alpha_0,\beta_0$ must be careful, and we choose the uniform distribution due to the above reasons. We have **provided a detailed explanation of this in Appendix B.3** of the revised paper.
>
> ---
>
> > Q7. Is variational inference computationally expensive, and does it risk converging to a local optimum?
>
> **A7.** Thank you for raising these important concerns. In this work, we employ efficient techniques, such as the reparameterization trick to enhance the scalability. In practice, the computational cost of training the Beta model via variational inference is comparable to training the reward model, **adding only a minimal amount of time** (several minutes) to the overall training process, whereas the subsequent RL training steps are more computationally demanding (taking several hours).
>
> Regarding the risk of converging to a local optimum, it is true that variational inference with neural networks, like many deep learning techniques, might converge to local optima. However, we mitigate this risk by incorporating techniques like dropout layers, into our neural network model. Additionally, the variational objective we optimize is generally well-behaved in practice, and our experiments have shown that it converges to a reasonable solution with a stable, small final loss.
>
> ---
>
> > Q8. The authors refer to two prior works on PbRL that especially focus on confidence and uncertainty [1,2], which could serve as reasonable baselines. Why were these approaches not included for comparison?
>
> **A8.** Thanks for raising the concern. Indeed, the two referenced prior works both address the **online PbRL setting**, where agents actively interact with the environment, query for preferences, and continuously update the reward model and policy based on newly acquired preference data. In contrast, our paper focuses on the **offline PbRL setting**, where learning is conducted solely on a fixed-size offline dataset, without any interaction with the environment or access to new preference labels. In this setting, the reward model is trained on the entire offline dataset and subsequently used to guide offline RL without retraining.
>
> More importantly, the two referenced papers [1,2] address the challenge of learning from diverse human preferences, which may be inconsistent or noisy. The uncertainty arising from such inconsistencies is attributed to **aleatoric uncertainty** [3,4], which reflects the **inherent variability or noise in the data**. In contrast, our paper focuses primarily on **epistemic uncertainty**, which emerges in offline datasets due to the limited comparison numbers. This type of uncertainty stems from a **lack of sufficient data**. Our work focuses on learning from a fixed-size offline dataset and managing the epistemic uncertainty it introduces. Consequently, the central focus of our modeling is on epistemic uncertainty, while aleatoric uncertainty lies beyond the scope of this paper.
>
> ---
>
> > Q9. The uncertainty of preference alignment in LLMs may have different motivations compared to control tasks. A brief discussion on this distinction in the introduction or experiment section would be beneficial.
>
> **A9.** Thanks for your suggestion. We have included such a discussion in the experiment section (Section 6.4) in the revised paper.

---

> ### Author Response · Authors · 2024-11-21
> **Author response to Reviewer oXZj - Part 3**
>
> > Q10. The paper lacks an evaluation on offline dataset collected from real human feedback.
>
> **A10.** Thanks for your valuable comments. In this work, we primarily use synthetic preference data, similar to many offline PbRL studies, which also generate synthetic preferences because of its convenience and effectiveness [5-7]. We believe that synthetic labels can also represent preferences to some extent.
>
> However, we recognize the importance of evaluating the methods with real human preferences, and we have **conducted additional experiments with real human preferences**. Specifically, we conduct experiments on risk-sensitive D4RL environments [8] with public real human preferences from [9].
>
> **The experiment details and results can be found in Appendix D.3 in the revised paper**. We find that UA-PbRL generally outperforms other baselines across the four tasks. This advantage stems from the inherent sparsity of riskier behaviors in the offline datasets, which introduces greater uncertainty in reward modeling. UA-PbRL effectively captures this uncertainty, helping it avoid such risky behaviors even if they may be preferred and sometimes yield higher rewards.
>
> ---
> **References**
>
> [1] Xue, Wanqi, et al. "Reinforcement learning from diverse human preferences." arXiv preprint arXiv:2301.11774 (2023).
>
> [2] Cheng, Jie, et al. "RIME: Robust Preference-based Reinforcement Learning with Noisy Preferences." ICML (2024).
>
> [3] Smith, Lewis, and Yarin Gal. "Understanding measures of uncertainty for adversarial example detection." UAI (2018).
>
> [4] Hüllermeier, Eyke, and Willem Waegeman. "Aleatoric and epistemic uncertainty in machine learning: An introduction to concepts and methods." Machine learning (2021).
>
> [5] Lee, Kimin, et al. "B-pref: Benchmarking preference-based reinforcement learning." NeurIPS (2021).
>
> [6] Shin, Daniel, et al. "Benchmarks and algorithms for offline preference-based reward learning." TMLR (2023).
>
> [7] Choi, Heewoong, et al. "Listwise reward estimation for offline preference-based reinforcement learning." ICML (2024).
>
> [8] Urpí, Núria Armengol, et al. "Risk-averse offline reinforcement learning." ICLR (2021).
>
> [9] Yuan, Yifu, et al. "Uni-RLHF: Universal Platform and Benchmark Suite for Reinforcement Learning with Diverse Human Feedback." ICLR (2024).

---

> ### Author Response · Authors · 2024-11-27
> **Kind reminder regarding your feedback**
>
> Dear Reviewer oXZj,
>
> We hope this message finds you well. We understand this might be a particularly busy time for you, and we genuinely appreciate the time you dedicate to reviewing.
>
> We would like to kindly remind you to review our rebuttal at your convenience. Your valuable feedback is instrumental in helping us refine and improve our paper. Should there be any specific points that require further clarification, please do not hesitate to let us know.
>
> Thank you for your time, effort, and thoughtful insights. We sincerely appreciate your contributions to this process.
>
> Best regards,
>
> Authors of Paper 6083

---

> ### Author Response · Authors · 2024-12-02
> **Kind reminder regarding your feedback on ICLR Paper 6083**
>
> Dear Reviewer oXZj,
>
> We hope this message finds you well. We understand and respect the many responsibilities you manage, and we greatly appreciate your time and effort in reviewing our paper.
>
> As the discussion period is nearing its conclusion, we wanted to kindly follow up regarding the feedback of our rebuttal for ICLR Paper 6083. Your feedback is invaluable to us, and we are eager to address any additional questions or concerns you may have.
>
> We deeply appreciate your contribution to this process. Thank you again for your time and thoughtful review.
>
> Best regards,
>
> Authors of Paper 6083

---

> ### Comment · Reviewer_oXZj · 2024-12-02
>
> I sincerely apologize for the delayed response. I greatly appreciate the authors' detailed replies and the revised manuscripts, which addressed my concerns. I will be raising my score accordingly.

---

> > ### Author Response · Authors · 2024-12-02
> >
> > We sincerely value your thoughtful and constructive feedback and deeply appreciate the time and effort you dedicated to reviewing our work. Your insightful comments have been invaluable in guiding us to improve the quality and clarity of our paper. Thank you very much!

---

### Official Review · Reviewer_TycP · 2024-11-03

**Soundness:** 3
**Presentation:** 3
**Contribution:** 3
**Rating:** 6
**Confidence:** 3

**Summary:**

The paper introduces Uncertainty-Aware Preference-based Reinforcement Learning (UA-PbRL), a framework for learning policies from offline human preference data while accounting for uncertainties in trajectory comparisons.
UA-PbRL proposes to first train an informative prior, followed by distributional reward model with a Maximum A Posteriori (MAP) objective to improve reward estimation and policy updates in the presence of uncertain samples.
By employing the Conditional Value-at-Risk (CVaR) metric, UA-PbRL facilitates risk-sensitive policy learning, demonstrating enhanced risk-averse behaviors across tasks like robot control and language model alignment.

**Strengths:**

1. The motivation for the need to learn a distributional reward function through offline preference data is well understood.
2. The methodology for learning the distributional reward function is theoretically well-grounded and novel.
3. The effectiveness of the UA-PbRL framework is demonstrated through experiments across various domains (from simple gridworld to LLM alignment).

**Weaknesses:**

1. Notation issues
* In Equation 9, an explanation of the distance metric is omitted, and additional question about the distance metric is detailed in Q4.
* In line 318, isn't Zπ a random variable? This seems to be expressed as an expectation.
2. Experiment on LLM alignment
* In the LLM alignment experiment, the main focus have to be results about red teaming experiment (i.e., how uncertain inputs are handled). However, there is a lack of comparative analysis on risk-averse behaivor of LLMs given harmful prompts that were not seen during the reward model training process.
* It would be helpful to provide information on how much the rate of evasive responses increases compared to the baseline (currently, only an example for one sample is provided in Appendix D.3).
3. Analysis on the various preference labelling setup.
* Over entire experiments, expected likelihood of a risky trajectory outranking a risk-averse one is set to be 0.6.
* At least in the gridworld experiment, I want to see how the results change with preference labelling setup other than p(τ2 >τ1 )=0.6.
* Specifically, it would be useful to analyze whether the uncertainty-aware reward function still estimates high variance for τ2  even when the preference for τ2  increases to around 0.7 or 0.8.

**Questions:**

1. Can you provide additional explanation for Equation 2 (derivation of the prior of the reward function)? I have difficulty understanding the transition from the first term to the second term and from the third term to the fourth term (Does d indicate a derivative?).

2. Rather than training neural network for informative beta prior, isn’t it possible to simply define parameters of Beta distribution using the statistics in the data (i.e., the number of times a specific trajectory wins over another trajectory / loses within the offline preference data)?  Of course, if you aimed to obtain the distribution of ϕ(τ) for any arbitrary trajectory, then this approach might be necessary, but aren’t you using offline trajectories D to train the reward model?

3. Regarding the proof of Theorem 4.1, Is there a guarantee that the p.d.f of the posterior distribution of r(τ) is concave? If not, how can Equation 22 ensure a “global” maximum?

4. r^K in the equation 9, which is a point estimate of the posterior distribution, seems to serve as a target value for the distributional reward model. But I wonder how does it provide sufficient learning signal for the distributional reward model to output correct \sigma, as the r^K does not contain any information about the variance of the posterior distribution.

---

> ### Author Response · Authors · 2024-11-21
> **Author response to Reviewer TycP - Part 1**
>
> Dear Reviewer, we sincerely value your time and effort in evaluating our work. We have prepared comprehensive responses and clarifications to address each point you raised. We hope these responses can resolve your concerns.
>
>
> > Comment 1. In line 318, isn't $Z^\pi$ a random variable? This seems to be expressed as an expectation.
>
> **Response 1.** $Z^\pi$ is a random variable, and we utilize Dirac functions to model its quantiles. Specifically, we use uniformly distributed Dirac functions to model the inverse cumulative density function (quantiles) of the distribution of $Z^\pi$. This is a common method in quantile regression distributional RL.
> Please refer to Equation 7 in [1] for more details.
>
> ---
>
> > Comment 2. In the LLM alignment experiment, the main focus have to be results about red teaming experiment (i.e., how uncertain inputs are handled). However, there is a lack of comparative analysis on risk-averse behaivor of LLMs given harmful prompts that were not seen during the reward model training process.
>
> **Response 2.** Thank you for raising the point. In our experiment, the harmful test prompts are indeed drawn from a test set that is separate from the training prompt-response pairs, ensuring that the reward model has not encountered these prompts during training.
>
> ---
>
> > Comment 3. It would be helpful to provide information on how much the rate of evasive responses increases compared to the baseline (currently, only an example for one sample is provided in Appendix D.3).
>
> **Response 3.** Thank you for your valuable suggestion. In the revised paper, we have included quantitative results on the increase in evasive responses compared to the baseline, by assessing the harmlessness of the generated responses to harmful prompts using prompted GPT-4o. Please refer to **Table 6 in Appendix D.5 for detailed results**. The results demonstrate that UA-PbRL generates more harmless responses than other methods.
>
> ---
>
> > Comment 4. Over entire experiments, expected likelihood of a risky trajectory outranking a risk-averse one is set to be 0.6. At least in the gridworld experiment, I want to see how the results change with preference labelling setup other than p($\tau$2 $>$ $\tau$1 )=0.6. Specifically, it would be useful to analyze whether the uncertainty-aware reward function still estimates high variance for $\tau$2 even when the preference for $\tau$2 increases to around 0.7 or 0.8.
>
> **Response 4.** Thank you for your valuable suggestion. We appreciate that adjusting the preference labeling probability and analyzing the results could yield additional insights.
>
> In response, we have conducted experiments across three Gridworld environments with **preference labeling probabilities of p=0.7 and 0.8**. The results for each Gridworld are presented in **Figures 9, 10, and 11**, respectively, in **Appendix D.1**.
>
> The results indicate that as the preference strength for risky ones increases, the difference in expectation grows while the variance difference remains relatively stable. This highlights that the uncertainty captured by the distributional reward primarily stems from the offline dataset.
>
> ---
>
> > Comment 5. Can you provide additional explanation for Equation 2 (derivation of the prior of the reward function)? I have difficulty understanding the transition from the first term to the second term and from the third term to the fourth term (Does d indicate a derivative?).
>
> **Response 5.** Sorry for the misunderstanding. The notation $d$ refers to the derivative operation. We think maybe you missed the definition of $\phi$, as defined in Line 223. Specifically, $\phi(\tau) = \frac{e^{r_\varphi(\tau)}}{e^{r_\varphi(\tau)} + 1}$ (we use $\phi$ as shorthand). By applying the chain rule, we can derive the transition from the first to the second term. As for the third to the fourth term, the deviation is just applied with the defined $\phi$ to get the results.
>
> ---
>
> > Comment 6. Rather than training neural network for informative beta prior, isn’t it possible to simply define parameters of Beta distribution using the statistics in the data (i.e., the number of times a specific trajectory wins over another trajectory / loses within the offline preference data)?
>
> **Response 6.** Thank you for raising this concern. You are right. In our implementation, we included an option to obtain a Beta distribution derived from data statistics without neural networks, as you suggested. This approach was specifically applied in the Gridworld environment, where the state-action space is discrete and finite. Specifically, we simulate the "voting" process as described at the beginning of Section 4.2 to get the Beta distribution driven by statistics.
> For continuous environments where trajectories are infinite, we opted to train a neural network, as described in the paper, to handle the complexities of these settings better.
>
> We have **incorporated this description into Appendix B.4** in the revised paper.

---

> ### Author Response · Authors · 2024-11-21
> **Author response to Reviewer TycP - Part 2**
>
> > Comment 7. Regarding the proof of Theorem 4.1, Is there a guarantee that the p.d.f of the posterior distribution of $r(\tau)$ is concave? If not, how can Equation 22 ensure a “global” maximum?
>
> **Response 7.** Thank you for bringing up this concern. It has been shown in [2,3] that the likelihood function of  $r(\tau)$ has a unique stationary point that is also a global maximum, given $e^r \geq 0$, under the conditions that 1) the geometric mean of the strengths is one (as enforced in our paper), and 2) the network adjacency matrix $ \omega^{i,j} $ is strongly connected. Based on this, [4] extends the results to the posterior function, proving that a global maximum exists without needing the aforementioned second condition. Therefore, as demonstrated in [4], the posterior distribution with a geometric mean of strengths equal to one guarantees a global maximum.
>
> ---
>
> > Comment 8. In Equation 9, an explanation of the distance metric is omitted... $r^K$ in the equation 9, which is a point estimate of the posterior distribution, seems to serve as a target value for the distributional reward model. But I wonder how does it provide sufficient learning signal for the distributional reward model to output correct $\sigma$, as the $r^K$ does not contain any information about the variance of the posterior distribution.
>
> **Response 8.** Thanks for the question. In Equation 9, dist() refers to the distance metric, for which we use the Mean Squared Error (MSE).
>
> You are correct that $r^K$ serves as a point estimate of the target value for the distributional reward model. In practice, we sample $N$ predicted rewards $\hat{r}^0$ from the reward model and calculate the corresponding $N$ target values $r^K$ for each trajectory. These values are then utilized for updating, providing a richer learning signal about the posterior distribution.
>
> ---
> **Referemces**
>
> [1] Dabney, Will, et al. "Distributional reinforcement learning with quantile regression." AAAI (2018).
>
> [2] Zermelo, Ernst. "Die berechnung der turnier-ergebnisse als ein maximumproblem der wahrscheinlichkeitsrechnung." Mathematische Zeitschrift (1929).
>
> [3] Ford Jr, Lester R. "Solution of a ranking problem from binary comparisons." The American Mathematical Monthly (1957).
>
> [4] Newman, M. E. J. "Efficient computation of rankings from pairwise comparisons." JMLR (2023).

---

> > ### Comment · Reviewer_TycP · 2024-11-25
> >
> > Thank you for the detailed response.
> >
> > [Comment 7]. I understood the point, and it would be better if you include such a contents in your manuscript.
> >
> > [Comment 8]. I am still not sure about your answer. Even if UA-pbRL computes N numbers of target values, they are all estimates of MAP (as far as I understood), so I am not sure how can such estimates provide learning signal for "Variance" of the posterior.
> > If there is something I miss, please correct me.
> >
> > [Comment 6]. Regarding this point, It would be better to include this version of UA-pbRL (using Beta distribution parameterized by data statistics) in your final draft. Additionally, can you explain more about the reason why you opt to use neural network in non-episodic tasks?

---

> > > ### Author Response · Authors · 2024-11-25
> > > **Authors' further response to Reviewer TycP**
> > >
> > > Dear Reviewer, thank you for evaluating our work and for your insightful feedback throughout the discussion process. We appreciate the time and efforts you have dedicated to reviewing our work. We hope the following response will effectively address your remaining concerns.
> > >
> > > > [Comment 7]. I understood the point, and it would be better if you include such a contents in your manuscript.
> > >
> > > **Response.** Thanks for your valuable suggestion. We have included such a discussion in the proof part of the revised paper (Appendix A).
> > >
> > > ---
> > >
> > > > [Comment 8]. I am still not sure about your answer. Even if UA-pbRL computes N numbers of target values, they are all estimates of MAP (as far as I understood), so I am not sure how can such estimates provide learning signal for "Variance" of the posterior. If there is something I miss, please correct me.
> > >
> > > **Response.** Sorry for the misunderstanding. We provide a more detailed description here.
> > >
> > > - On the one hand, in practice, for varying initial reward values $\hat{r}_0$, the target values $\hat{r}^K$ computed after a fixed number of $K$ iterations may not precisely correspond to the optimal solution of the MAP objective. Nevertheless, as each iteration progressively approaches the MAP objective, the resulting estimates represent multiple distinct points within the underlying posterior. The variance among these estimates can provide information for distributional reward learning (i.e., learning $\sigma$).
> > >
> > > - On the other hand, this variance ($\sigma$) is inherently tied to the variance of the posterior because, for different trajectories, the iteration objective varies with different values of $\alpha$ and $\beta$. Consequently, the variance among $N$ estimates $\hat{r}^K$ for different trajectories will certainly differ. Intuitively, larger $\alpha$ and $\beta$ result in a stronger initial belief weight during the iteration, reducing the influence of the observed preference labels, thus with a smaller variance of the $N$ estimates.
> > >
> > > In summary, $\sigma$ can capture the information provided by the posterior variance through the $K$ iterations applied to $N$ sampled values. We have clarified these details in Appendix B.1 in the revised paper.
> > >
> > > ---
> > >
> > > > [Comment 6]. Regarding this point, It would be better to include this version of UA-pbRL (using Beta distribution parameterized by data statistics) in your final draft. Additionally, can you explain more about the reason why you opt to use neural network in non-episodic tasks?
> > >
> > > **Response.** Thanks for your valuable comments. We have included the discrete version of UA-PbRL mentioned in Section 4.2 and detailed in Appendix B.4, and we ensure clarity in the final draft.
> > >
> > > For neural networks, we primarily utilize them in continuous state-action spaces. This is because, in such spaces, trajectories will not exactly match, even if they are very close. This property makes it difficult and inappropriate to rely solely on data statistics. Using data statistics in this context would result in only a single comparison per trajectory, as the trajectories are distinct. Moreover, this approach fails to generalize to new trajectories that differ from those in the offline dataset. By contrast, neural networks are well-suited for modeling the Beta distribution, which can inherently capture similarities across trajectories, enabling more reasonable predictions. Furthermore, they can generalize to unseen trajectories, which is crucial for extending the model's applicability.
> > >
> > > ---
> > >
> > > We hope the above response could resolve your concerns. If you have more questions, please feel free to discuss them with us. Thank you once again for your invaluable feedback and the thoughtful effort you invested in reviewing our paper!

---

> > > > ### Comment · Reviewer_TycP · 2024-12-03
> > > >
> > > > Thank you for the detailed explanation.
> > > > My concerns have been resolved, and I will maintain the current recommendation as 6.
> > > >
> > > > Many thanks,
> > > > Reviewer TycP.

---

> > > > > ### Author Response · Authors · 2024-12-03
> > > > >
> > > > > Thank you for your acknowledgment and confirmation. We are truly pleased to hear that the explanation successfully addressed your concerns. Your thoughtful and detailed feedback has been invaluable in refining our work, and we greatly appreciate the time and effort you dedicated to reviewing our paper. Thank you!

---

### Official Review · Reviewer_hsS4 · 2024-11-04

**Soundness:** 2
**Presentation:** 2
**Contribution:** 2
**Rating:** 8
**Confidence:** 4

**Summary:**

This paper introduces Uncertainty-Aware Preference-Based Reinforcement Learning (UA-PbRL), an uncertainty-aware preference alignment approach for PbRL. Unlike traditional PbRL approaches that are often risk-neutral, UA-PbRL leverages a distributional reward model and a risk-sensitive policy to better capture uncertainties inherent in human preferences. Key components include a Maximum A Posteriori (MAP) objective for reward updates using an informative Beta prior and a Conditional Value-at-Risk (CVaR) metric to develop a risk-averse policy. The method is evaluated across various domains, such as robot control, pointmaze, and language model alignment tasks, demonstrating its ability to handle uncertain scenarios effectively.

**Strengths:**

- The use of a distributional approach and risk-sensitive metrics provides a robust framework for handling uncertainty in human preference data.
- The empirical results demonstrate that UA-PbRL outperforms baseline methods in both mean and worst-case performance, validating its efficacy in high-risk scenarios.
- The empirical results across multiple domains (robot navigation, LLM alignment) highlights its adaptability for wide adoption.
- The code is provided for reproducibility.

**Weaknesses:**

- Although the method is evaluated across various domains, the evaluation on stochastic D4RL [1], which is an important benchmark for risk-averse offline RL (used in both O-RAAC and CODAC), is missing.
- Lack of discussing recent online PbRL methods [2, 3, 4, 5].
- The LLM experiments lack proper baseline. If the authors want to show the effectiveness of UA-PbRL in the LLM setting, they should compare it with the state-of-the-art methods in the LLM setting, such as [6].
- The choice of baselines could be further discussed, especially regarding differences in reward model architectures. If Transformer-based models are not used consistently, this may influence performance comparisons.

References:

[1] Risk-averse offline reinforcement learning. ICLR 2021.

[2] Meta-Reward-Net: Implicitly Differentiable Reward Learning for Preference-based Reinforcement Learning. NeurIPS 2022.

[3] Few-Shot Preference Learning for Human-in-the-Loop RL. CoRL 2023.

[4] Hindsight PRIORs for Reward Learning from Human Preferences. ICLR 2024.

[5] PIPER: Primitive-Informed Preference-based Hierarchical Reinforcement Learning via Hindsight Relabeling. ICML 2024.

[6] Distributional Preference Learning: Understanding and Accounting for Hidden Context in RLHF. ICLR 2024.

**Questions:**

Besides the weaknesses above, further question is as follows:

- Whether the baselines use Transformer-based reward models? If not, I think the comparison is not fair. At least, the authors should compare with Preference Transformer to ignore the architecture difference.
- Since UA-PbRL’s policy optimization is risk-averse, how does this impact its overall performance, particularly in tasks where risk-neutral policies might perform better in terms of mean rewards?
- How does UA-PbRL perform when the human preference data is noisy or inconsistent?

---

> ### Author Response · Authors · 2024-11-21
> **Author response to Reviewer hsS4 - Part 1**
>
> Dear Reviewer, we sincerely appreciate your valuable and constructive comments. We have carefully considered each of your questions, and we hope that the following responses address your concerns satisfactorily.
>
> > Comment 1. Although the method is evaluated across various domains, the evaluation on stochastic D4RL [1], which is an important benchmark for risk-averse offline RL (used in both O-RAAC and CODAC), is missing.
>
> **Response 1.** Thank you for your thoughtful insights. We appreciate the importance of the stochastic (i.e., risk-sensitive) D4RL [1] setup. In the revised paper, we have generalized the risk-sensitive D4RL tasks to the offline PbRL setting with human preferences provided by [2].
>
> **The experiment details and results can be found in Appendix D.3**. We find that UA-PbRL generally outperforms other baselines across the four tasks. This advantage stems from the **inherent sparsity of riskier behaviors** in the offline datasets, which introduces greater uncertainty in reward modeling. UA-PbRL effectively captures this uncertainty, helping it avoid such risky behaviors even if they may be preferred and sometimes yield higher rewards.
>
> ---
>
> > Comment 2. Lack of discussing recent online PbRL methods.
>
> **Response 2.** Thanks for raising this concern. This paper focuses on the **offline setting**, and we have discussed the most relevant online PbRL works in the introduction and related works. Nevertheless, we recognize the value of discussing more recent online PbRL methods. In our revised manuscript, we have discussed these online PbRL works you referred to in Section 2 marked by blue.
>
> ---
>
> > Comment 3. The LLM experiments lack proper baseline. If the authors want to show the effectiveness of UA-PbRL in the LLM setting, they should compare it with the state-of-the-art methods in the LLM setting, such as [6].
>
> **Response 3.** Thank you for bringing this paper to attention. Distributional Preference Learning (DPL) [3] considers learning a reward distribution that accounts for hidden contexts, such as combined preference data with varied objectives. However, our concern is whether DRL can serve as a proper baseline since DPL authors argue in their paper that **"DPL is not trying to model **uncertainty which comes from limited data**, but rather **uncertainty which comes from hidden context**."** Indeed, our paper specifically focuses on the former one—uncertainty arising from limited preference data.
>
> Despite the above concern, to better address your questions and include state-of-the-art methods, we **conduct additional experiments** to include the DPL method for preference alignment. **The detailed experimental setup, results, and analysis are provided in Section 6.4 and Appendix D.5**.
>
> ---
>
> > Comment 4. The choice of baselines could be further discussed, especially regarding differences in reward model architectures. If Transformer-based models are not used consistently, this may influence performance comparisons. Whether the baselines use Transformer-based reward models? If not, I think the comparison is not fair. At least, the authors should compare with Preference Transformer to ignore the architecture difference.
>
> **Response 4.** We appreciate this suggestion, but our baselines indeed utilize the Transformer architectures. We have already clarified in the submitted paper that all methods are built upon the Preference Transformer architecture to ensure a fair comparison. This is noted in Section 6, Line 381, and in Appendix B.3, Line 945.
>
> ---
>
> > Comment 5. Since UA-PbRL’s policy optimization is risk-averse, how does this impact its overall performance, particularly in tasks where risk-neutral policies might perform better in terms of mean rewards?
>
> **Response 5.** Thanks for your comments.
> We must first clarify that the main focus of our work is to seek a policy that prioritizes safety and reliability by avoiding risky or high-uncertainty states (e.g., environments with stochastic dynamics or limited observation).
>
> We recognize that a risk-averse policy may not always outperform risk-neutral policies in terms of mean rewards. This reflects a trade-off between performance and safety, which is primarily determined by the risk level $\alpha$ during policy optimization, representing the degree of risk aversion. Intuitively, smaller values of $\alpha$ correspond to higher risk aversion, reducing the likelihood of violations. However, this increased risk aversion can also negatively affect reward performance.
>
> To solve your concern, we perform **an ablation study on the Risky Ant environment with varying $\alpha$ values**. The results and analysis can be found in **Table 5 in Appendix D.4.**

---

> ### Author Response · Authors · 2024-11-21
> **Author response to Reviewer hsS4 - Part 2**
>
> > Comment 6. How does UA-PbRL perform when the human preference data is noisy or inconsistent?
>
> **Response 6.** Thanks for raising this concern. We would like to clarify that this paper primarily addresses the uncertainty arising from imbalanced offline datasets, where the sparsity of compared trajectories leads to increased uncertainty in the learned reward model. This type of uncertainty is classified as **epistemic uncertainty**, which stems from **a lack of sufficient data**. Our approach focuses on managing this epistemic uncertainty by incorporating priors and updating the reward model through a MAP estimation.
>
> In contrast, we do not focus on the consistency of human preferences, which represents a different form of uncertainty known as **aleatoric uncertainty**, which refers to the **inherent variability or noise in the data**. This type of uncertainty is not within the scope of our paper. For more details, please refer to [4,5].
>
> ---
> **References**
>
> [1] Urpí, Núria Armengol, et al. "Risk-averse offline reinforcement learning." ICLR (2021).
>
> [2] Yuan, Yifu, et al. "Uni-RLHF: Universal Platform and Benchmark Suite for Reinforcement Learning with Diverse Human Feedback." ICLR (2024).
>
> [3] Siththaranjan, Anand, et al. "Distributional preference learning: Understanding and accounting for hidden context in RLHF." ICLR (2024).
>
> [4] Smith, Lewis, and Yarin Gal. "Understanding measures of uncertainty for adversarial example detection." UAI (2018).
>
> [5] Hüllermeier, Eyke, and Willem Waegeman. "Aleatoric and epistemic uncertainty in machine learning: An introduction to concepts and methods." Machine learning (2021).

---

> > ### Comment · Reviewer_hsS4 · 2024-11-24
> > **Further response**
> >
> > Thank you for the detailed response and the new empirical results. In Response 6 and A8 to Reviewer oXZj, the authors claim that this paper primarily addresses the uncertainty arising from insufficient data, classifying it as epistemic uncertainty. However, the experiments are conducted in Gridworld, Risky PointMaze, and Risky Robot Control by introducing a risky region where environmental transitions are perturbed with additional Gaussian noise and the trajectories are sampled with a uniform probability. I believe this experimental setting does not fully represent the situation of insufficient data. Regarding uniform sampling, while there is a probability that the frequency of sampled trajectory pairs may be highly imbalanced, I am still concerned that the sampled pairs are uniform, i.e., at least $90\\%$ trajectories are sampled once in the preference dataset. I suggest that the authors include statistics of the sampling process as a motivating experiment to address aleatoric uncertainty in offline PbRL. I am satisfied with other responses and decide to change my score to 5.

---

> ### Author Response · Authors · 2024-11-25
> **Authors' further response to Review hsS4**
>
> Dear Reviewer, we sincerely appreciate your constructive feedback and are grateful for the time and effort you've invested in reviewing our work. We hope the following response can address your remaining concerns in two points.
>
> > 1. The authors claim that this paper primarily addresses the uncertainty arising from insufficient data, classifying it as epistemic uncertainty. However, the experiments are conducted in Gridworld, Risky PointMaze, and Risky Robot Control by introducing a risky region where environmental transitions are perturbed with additional Gaussian noise and the trajectories are sampled with a uniform probability.
>
> **Response 1.** Thanks for raising this concern. In the offline PbRL setting, the term "insufficient data" specifically refers to **"insufficient preference label corresponding with certain trajectories."** In our paper, we follow the standard literature to define epistemic uncertainty. By definition, epistemic uncertainty is "uncertainty due to our lack of knowledge; we are uncertain because we lack understanding." This occurs when model parameters are not well-defined because of **limited data** [1].
>
> Regarding the experiments, **our experiments are designed to explain why epidemic uncertainty will arise from the preference dataset**. For example, **in risky robot control tasks**, to simulate such cases, we include risky regions and perform uniform sampling to construct the preference dataset. In this dataset, **trajectories with fewer comparisons induce significant epistemic uncertainty, which arises from the stochastic environmental transitions**. Such limited comparisons prevent reliable determination of their quality. These fewer-compared trajectories are indeed "risky." In the **risk-sensitive D4RL tasks, risky behaviors (e.g., high velocity) are indeed less frequent in the provided offline dataset.** Under the uniform sampling procedure, this scarcity naturally results in fewer comparisons and introduces greater uncertainty. By identifying such uncertainty and implementing a risk-averse policy, the model can effectively avoid generating risky behaviors.
>
> In essence, some of the confusion arises from the shift of perspective:
>
> - From the perspective of the sampling policy, aleatoric uncertainty in environmental transitions affects the policies, **resulting in different levels of diversity** and influencing the number of samples in the generated dataset.
>
> - When we transfer to the perspective of data-driven offline PbRL, it is **the diversity of these samples that induces varying degrees of comparison**, which in turn leads to different levels of epistemic uncertainty.
>
> [1] Smith, Lewis, and Yarin Gal. "Understanding measures of uncertainty for adversarial example detection." UAI (2018).
>
> ---
>
> > 2. I believe this experimental setting does not fully represent the situation of insufficient data. Regarding uniform sampling, while there is a probability that the frequency of sampled trajectory pairs may be highly imbalanced, I am still concerned that the sampled pairs are uniform, i.e., at least $90\%$ trajectories are sampled once in the preference dataset. I suggest that the authors include statistics of the sampling process as a motivating experiment to address aleatoric uncertainty in offline PbRL.
>
> **Response 2.** Thanks for this question, and sorry for the misunderstanding about the sampled trajectory pairs. Indeed, **our experimental setting does simulate the situation of insufficient (or imbalanced) data**. We adopt a uniform sampling procedure that generates a substantial number of comparisons across various trajectory segments. **The total steps involved in these sampled comparison trajectory segments significantly exceed the total steps of the original trajectories**, resulting in an imbalanced preference dataset where only few trajectories are sampled only once. The details can be found in **Appendix B.2** in the revised paper.
>
> We have calculated the distribution that records the frequency at which trajectory segments are sampled and compared. **The relevant statistics are presented in the table below**, which demonstrates the divergence in the offline preference dataset. In continuous environments, directly calculating the number of comparison trajectories is challenging. However, the same underlying principle applies, and neural networks can effectively identify patterns if similar trajectories are compared more frequently.
>
> ||min|10\% quantile|90\% quantile|99\% quantile|max|
> |:---:|:---:|:---:|:---:|:---:|:---:|
> |Gridworld 1|1|3|11|94|1277|
> |Gridworld 2|1|5|13|225|1011|
> |Gridworld 3|1|5|12|166|617|
>
> Once again, we emphasize that our paper focuses on the epistemic uncertainty arising from the offline preference dataset.
>
> ---
>
> We hope the above response could resolve your concerns. If you have more questions, please feel free to discuss them with us. Thank you once again for your invaluable feedback and the thoughtful effort you invested in reviewing our paper!

---

> ### Author Response · Authors · 2024-12-02
> **Kind reminder regarding your feedback on ICLR Paper 6083**
>
> Dear Reviewer hsS4,
>
> We hope this message finds you well. Thank you once again for your thoughtful feedback and for the additional comments regarding our rebuttal.
>
> We have carefully considered your comments and provided detailed clarifications addressing the points you raised. We truly appreciate your insights and the opportunity to further refine our work.
>
> At your convenience, we kindly ask if you could review our further responses to ensure that we have addressed your concerns adequately.
>
> Thank you for your time and dedication to this process. Your input is incredibly valuable to us.
>
> Best regards,
>
> Authors of Paper 6083

---

> ### Comment · Reviewer_hsS4 · 2024-12-03
> **Thanks for the response**
>
> Thanks for the detailed response, and I apologize for the delay in providing my feedback. I appreciate the effort the authors have put into addressing my concerns. The explanation regarding epistemic uncertainty in the offline PbRL setting is clear. I believe this clarification strengthens the contribution of the paper. I would like to offer two suggestions for the future camera-ready version to further improve the presentation and impact of the work:
>
> 1. The paper would benefit from a stronger motivation in Section 1. Specifically, incorporating the explanation provided regarding epistemic uncertainty of the offline PbRL setting into this section would help better contextualize the problem. Additionally, including visualizations such as density curves of the preference data could make the setting more intuitive.
> 2. I recommend the authors including an ablation study to investigate how the performance of UA-PbRL changes under varying degrees of insufficient data. This analysis would provide deeper insights of the proposed method and assess the extent of uncertainty that UA-PbRL can effectively handle.
>
> In light of the improvements and the authors’ response, I have decided to raise my score to 8. Thank you again for your hard work and for addressing my concerns comprehensively.

---

> > ### Author Response · Authors · 2024-12-03
> > **Thanks for your reply and suggestions**
> >
> > Thank you for your acknowledgment and confirmation. We greatly appreciate your thoughtful and detailed feedback, which has played a crucial role in refining and improving our work. The time and effort you dedicated to thoroughly reviewing our paper are deeply valued, and your insights have been invaluable in shaping the direction of our revisions.
> >
> > We will carefully incorporate your valuable suggestions into the final version of the paper to further enhance its presentation and overall quality.
> >
> > Thank you once again for your support, guidance, and constructive feedback! Your contributions are truly appreciated and have significantly enriched our research.

---

### Official Review · Reviewer_daK1 · 2024-11-10

**Soundness:** 4
**Presentation:** 3
**Contribution:** 4
**Rating:** 8
**Confidence:** 4

**Summary:**

The paper proposes an uncertainty-aware version of the preference-based RL (PbRL) framework for offline settings. The method first fits a distributional model of human rewards using offline preference data using a MAP estimate with a learned beta prior. This model is then used with an offline distributional policy learning objective, enabling risk-averse optimization with CVaR. Didactic results are presented in a gridworld, as well as evaluation on control tasks (maze and swimmer), and in an LLM RLHF setting, showing an improved ability to safely adhere to human preferences.

**Strengths:**

- Better algorithms for risk-averse optimization of human preference models is an important and timely problem for creating safe and aligned AI agents
- Approach shows good performance across diverse settings, including in an LLM setting, where widely-adopted RLHF methods could benefit from methods to avoid catastrophic outcomes (see, e.g., Gemini safety training issues with racial bias)
- Didactic gridworld visualizations are helpful
- Algorithm is theoretically justified (see questions for a few points to clarify)

**Weaknesses:**

- All human preference data was synthetically generated
- A few points of confusion (see questions below)

**Questions:**

- What is dist() in Eq. (9)?
- Could the authors briefly discuss the advantages of the iterative approach for the MAP estimate in Eq. (8) / Theorem 4.1 over using a single step learned model?
- Similarly, what are the advantages of separately fitting the prior (Sec. 4.2) and then doing a MAP estimate (Sec. 4.3)? Is this performing a form of type II MLE? And could this be converted to a single step approach?
- Line 222: "we enforce the geometric mean strength to be one"; how is this enforced? Is this part of the prior in Eqs. (2) and (3), or a constraint?

Minor formatting:
- Line 277: "In specific" $\Rightarrow$ "To be specific"
- Line 245: Use ``...'' for quotes instead of '...'

---

> ### Author Response · Authors · 2024-11-21
> **Author response to Reviewer daK1 - Part 1**
>
> Dear Reviewer, we sincerely value your time and effort in evaluating our work, and thank you for your recognition. We have prepared comprehensive responses and clarifications to address each point you raised. We hope these responses can resolve your concerns.
>
> > Comment 1. All human preference data was synthetically generated.
>
> **Response 1.** Thank you for your comment. In this work, we primarily use synthetic preference data, similar to many offline PbRL studies, which also generate synthetic preferences because of its convenience and effectiveness [1-3]. We believe that synthetic labels can also represent preferences to some extent.
>
> However, we recognize the importance of evaluating the methods with real human preferences, and we have **conducted additional experiments with real human preferences**. Specifically, we conduct experiments on risk-sensitive D4RL environments [4] with public real human preferences from [5].
>
> **The experiment details and results can be found in Appendix D.3 in the revised paper**. We find that UA-PbRL generally outperforms other baselines across the four tasks. This advantage stems from the **inherent sparsity of riskier behaviors** in the offline datasets, which introduces greater uncertainty in reward modeling. UA-PbRL effectively captures this uncertainty, helping it avoid such risky behaviors even if they may be preferred and sometimes yield higher rewards.
>
> ---
>
> > Comment 2. What is dist() in Eq. (9)?
>
> **Response 2.** dist() represents the distance metric, for which we use the Mean Squared Error (MSE). We have revised the paper with a clearer notation.
>
> ---
>
> > Comment 3. Could the authors briefly discuss the advantages of the iterative approach for the MAP estimate in Eq. (8) / Theorem 4.1 over using a single step learned model?
>
> **Response 3.** Thank you for raising this insightful question. We assume that by the "iterative approach," the reviewer refers to the method where $K$ iterations are performed for computing the target value, followed by multiple updates to the reward model based on this target value. On the other hand, by the "single-step" method, the reviewer refers to the process where the reward model is updated based on only one single iteration's value, and this update is repeated until convergence. Both the iterative approach and single-step approaches can be effective in theory. However, "single-step" fitting induces a larger computational burden on reward model update (nearly $K$ times larger than the iterative approach). The burden is induced by the frequent update of the reward model at each iteration, which slows down our training.
>
> ---
>
> > Comment 4. Similarly, what are the advantages of separately fitting the prior (Sec. 4.2) and then doing a MAP estimate (Sec. 4.3)? Is this performing a form of type II MLE? And could this be converted to a single step approach?
>
> **Response 4.** Thanks for your question. The decision to separately fit the prior and then perform a MAP estimate is driven by different methods of learning prior and posterior. For the prior update, we utilize the ELBo objective to adapt the parameters of a neural network. For the posterior udpate (MAP), we implement the optimization via iterative update driven by our theoretical objective. Incorporating them into a single step requires careful consideration.
>
> In terms of type II MLE, its goal is to maximize the likelihood over both the model parameters and hyperparameters. We assume the reviewer is referring to estimating the model parameters while treating the prior distribution as part of the likelihood.
> Intuitively, our MAP shares relevant motivation with type II MLE, but there are some underlying distinctions. Specifically:
>
> - Our method involves a Bayesian update on the posterior distribution, while type II MLE does not. Indeed, our method is closely related to the **Empirical Bayesian** methods that involve data-driven estimation of prior and posterior.
>
> - As shown in the discussion above, our prior and posterior estimates can not be directly merged into a single-step update. Investigating the conversion to a single-step approach is an interesting direction for future work.

---

> ### Author Response · Authors · 2024-11-21
> **Author response to Reviewer daK1 - Part 2**
>
> > Comment 5. Line 222: "we enforce the geometric mean strength to be one"; how is this enforced? Is this part of the prior in Eqs. (2) and (3), or a constraint?
>
> **Response 5.** We enforce this by normalizing the rewards over the dataset to make $\sum_n r_n=0$, which indeed makes "geometric mean strength to be one." This normalization is feasible due to the following reasons:
>
> In essence, PbRL is solving a learning-to-rank problem, so the scale of strength ($e^r$) does not have a significant impact as long as the players' rankings are consistent.
>
> Specifically, the winning probability $p_{i,j}$ that trajectory $\tau_i$ with reward $r_i$ wins against trajectory $\tau_j$ with reward $r_j$ remains unchanged under multiplication of all the $e^{r_n}$ by any constant factor. So we can apply any convenient normalization condition. In this paper, following [6], we set the geometric mean strength to 1, which corresponds to $\prod_n{e^{r_n}}=1$ (i.e., $\sum_n r_n=0$). This choice has the beneficial effect that the probability $p$ of a trajectory with strength $e^{r}$ defeating the average trajectory (whose strength $e^{\bar{r}}=1$) is $p=\frac{e^r}{e^r+1}$, and thus $r=\log\frac{p}{1-p}$. Consequently, the reward parameter has a straightforward interpretation: it represents the log odds of defeating the average trajectory.
>
> ---
>
> > Comment 6. Minor formatting in Lines 277 and 245.
>
> **Response 6.** Thanks for your correction. We have updated them in the revised paper.
>
> ---
> **References**
>
> [1] Lee, Kimin, et al. "B-pref: Benchmarking preference-based reinforcement learning." NeurIPS (2021).
>
> [2] Shin, Daniel, et al. "Benchmarks and algorithms for offline preference-based reward learning." TMLR (2023).
>
> [3] Choi, Heewoong, et al. "Listwise reward estimation for offline preference-based reinforcement learning." ICML (2024).
>
> [4] Urpí, Núria Armengol, et al. "Risk-averse offline reinforcement learning." ICLR (2021).
>
> [5] Yuan, Yifu, et al. "Uni-RLHF: Universal Platform and Benchmark Suite for Reinforcement Learning with Diverse Human Feedback." ICLR (2024).
>
> [6] Newman, M. E. J. "Efficient computation of rankings from pairwise comparisons." JMLR (2023).

---

> > ### Comment · Reviewer_daK1 · 2024-12-03
> > **Response**
> >
> > Thank you for your response. I will keep my score of 8.

---

> > > ### Author Response · Authors · 2024-12-03
> > >
> > > Thank you for your recognition and confirmation! Your thoughtful and detailed review has been invaluable in helping us refine and improve our work. We deeply appreciate the time, effort, and expertise you brought to reviewing our paper. Thank you once again!

---

### Author Response · Authors · 2024-11-21
**Summary of updates**

We sincerely thank all the reviewers for their invaluable feedback and insightful suggestions, which have been instrumental in improving our work.

In response to the reviewers' comments, we incorporated detailed explanations, expanded discussions, and additional experimental results in the revised paper. The **modifications have been highlighted in blue** for clarity. We summarize our major updates below:

1. We **improved the clarity and readability** of the paper, including an explanation to Eq. (2), refinements to Eq. (9), and correcting some typos (Reviewers daK1, TycP, and oXZj).

2. We conducted **additional experiments utilizing real human preferences on risk-sensitive robot control tasks**, as detailed in Appendix D.3 (Reviewers daK1, hsS4, and oXZj).

3. We **added a discussion on recent online PbRL methods** in Section 2 (Reviewer hsS4).

4. We **integrated a state-of-the-art RLHF method** into LLM alignment experiments in Section 6.4 (Reviewer hsS4).

5. We **included an ablation study and a discussion on the risk-aversion capabilities** of the policy in Appendix D.4 (Reviewer hsS4).

6. We presented **quantitative results on evasive responses to harmful prompts** in LLM experiments in Appendix D.5 (Reviewer TycP).

7. We **explored variations in preference labeling probabilities** for risky and risk-averse trajectories, visualizing the resulting learned reward distributions in Appendix D.1 (Reviewer TycP).

8. We **discussed the initial belief in the Beta prior and its discrete implementation** in Appendix B.3 and Appendix B.4 (Reviewers TycP and oXZj).

We hope that our revisions can address the concerns raised and look forward to receiving some feedback from the reviewers. We are more than willing to engage in further discussions to refine our work.

---

### Meta-Review · Area_Chair_RUtd · 2024-12-18

**Metareview:**

**summary**

The paper introduces Uncertainty-Aware Preference-Based Reinforcement Learning (UA-PbRL), a framework for offline PbRL that incorporates uncertainty in both reward modeling and policy optimization. UA-PbRL employs a distributional reward model trained via a  MAP objective with a learned Beta prior, capturing uncertainties in human preferences. This model is combined with CVaR to enable risk-averse policy learning. Experiments across diverse tasks, such as gridworld, control tasks (maze and swimmer), and language model alignment, demonstrate UA-PbRL’s effectiveness in handling uncertain scenarios and outperforming baseline PbRL methods.

---

**strengths**

* Novel idea of introducing a distributional reward model with a MAP objective and leveraging risk-sensitive metrics like CVaR to optimize policies.
* Strong empirical performance: the authors demonstrate effectiveness across diverse tasks, including gridworld, robot control, and LLM alignment, highlighting its adaptability to various domains.
* Reproducibility and clarity: the paper is well-organized and the authors provide code for reproducibility, increasing trust and accessibility.

---

**weaknesses**

* Experiments were mainly based on synthetic preferences but the authors provided additional experiments with real human preferences during the rebuttal.
* Lack of proper baselines in LLM experiments.

---

**decision**

Overall, all reviewers agreed that this is a very solid submission and the authors also handled concerns from reviewers during the discussion period. I think the paper makes a nice contribution that the community will find valuable.

**Additional Comments On Reviewer Discussion:**

The authors addressed concerns from reviewers as follows:

* Improved clarity and readability
* Added experiments using real human preferences
* Discussed recent online PbRL methods
* Integrated a state-of-the-art RLHF method into LLM alignment experiments
* Added an ablation study and discussion on policy risk-aversion
* Presented quantitative results on harmful prompt responses in LLM experiments
* Explored preference labeling probabilities for risky vs. risk-averse trajectories
* Discussed the Beta prior's initial belief and its discrete implementation

---

### Decision · Program_Chairs · 2025-01-22

Accept (Poster)